

# Relationships between phytoplankton pigments and DNA- or RNA-based abundances support ecological applications

Robert H. Lampe[1,2], Ariel J. Rabines[1,2], Steffaney M. Wood[1,2,3], Anne Schulberg[1,2], Ralf Goericke[1], Pratap Venepally[2], Hong Zheng[2], Michael R. Stukel[4], Michael R. Landry[1], Andrew D. Barton[1,5], Andrew E. Allen[1,2]

[1]Integrative Oceanography Division, Scripps Institution of Oceanography, University of California San Diego, La Jolla, CA, 92093 USA
[2]Microbial and Environmental Genomics, J. Craig Venter Institute, La Jolla, CA, 92037, USA
[3]Center for Marine Biotechnology and Biomedicine, Scripps Institute of Oceanography, University of California San Diego, La Jolla, CA, 92093 USA
[4]Earth, Ocean, and Atmospheric Science Department, Florida State University, Tallahassee, FL, 32304 USA
[5]Department of Ecology, Behavior and Evolution, University of California San Diego, La Jolla, CA, 92093 USA

*Correspondence to*: Andrew E. Allen (aallen@ucsd.edu)

**Abstract.** Observations of phytoplankton abundances and community structure are critical towards understanding marine ecosystems. Common approaches to determine group-specific abundances include measuring phytoplankton pigments via high-performance liquid chromatography and DNA-based metabarcoding. Increasingly, mRNA abundances via metatranscriptomics are also employed. As phytoplankton pigments are used to develop and validate remote sensing algorithms, further comparisons between pigments and other metrics are needed to validate the extent to which these measurements agree for group-specific abundances; however, most previous comparisons have been hindered by metabarcoding and metatranscriptomics solely producing relative abundance data. By employing quantitative approaches that express both 18S rDNA and total mRNA as concentrations, we show that these measurements are related for several eukaryotic phytoplankton groups. We further propose that integration of these can be used to examine ecological patterns more deeply. For example, productivity-diversity relationships of both the whole community and individual groups show a dinoflagellate-driven negative trend rather than the commonly-found unimodal pattern. Pigments are also shown to relate to certain harmful algal bloom-forming taxa as well as the expression of sets of genes. Altogether, these results suggest that potential models of pigment concentrations via hyperspectral remote sensing may enable improved assessments of global phytoplankton community structure, including the detection of harmful algal blooms, and support the development of ecosystem models.

## 1 Introduction

Marine phytoplankton comprise both prokaryotic cyanobacteria and diverse lineages of eukaryotes with distinct evolutionary histories (Pierella Karlusich et al., 2020). Collectively, they are responsible for approximately 50% of global primary production, support marine food webs and extensively contribute to biogeochemical cycling (Huang et al., 2021). Their distinct



evolutionary trajectories have also given rise to different traits, functional roles, and niches. For example, diatoms (Bacillariophyta) tend to dominate under well-mixed, nutrient-rich conditions, are considered principal contributors to carbon export with fast sinking rates, and are a significant component of the silicon cycle due to their formation of silica cell walls

(Guidi et al., 2016; Agusti et al., 2015; Bowler et al., 2010) Dinoflagellates (Dinoflagellata) may be major contributors to carbon export in oligotrophic regions, and many species are mixotrophs, which enables them to serve multiple important roles in marine food webs (Guidi et al., 2016; Stoecker et al., 2017). Prymnesiophytes (Haptophyta) include coccolithophores with calcium carbonate shells, the formation of which influences upper ocean alkalinity and air-sea carbon dioxide exchange (Rost and Riebesell, 2004). The prymnesiophytes also include the bloom-forming *Phaeocystis spp.* which are a major source of the

sulfur compound, dimethylsulfoniopropanate (DMSP), that is hypothesized to influence cloud formation and climate (Smith Jr and Trimborn, 2023). Cryptophytes (Cryptophyta) are considered to be ubiquitous and may be highly abundant in the Sargasso Sea and Southern Ocean (Richardson, 2022; Cotti-Rausch et al., 2016; Mendes et al., 2018). Lastly, the chlorophytes (Chlorophyta) and pelagophytes (Pelagophyceae) are additional picoeukaryotes that are important primary producers, particularly in the open ocean (Worden et al., 2012; Not et al., 2012; Lopes dos Santos et al., 2017; Guérin et al., 2022).


Due to these contributions to ecosystem function and differing roles, assessing phytoplankton abundances and disentangling phytoplankton community composition (PCC) is necessary to understand marine environments (Cetinić et al., 2024). For example, current Earth systems models (ESMs) are unable to confidently project if primary productivity will increase or decrease under future climate scenarios (Kwiatkowski et al., 2020). By examining the effects of natural climate variability and

anthropogenic change on phytoplankton communities as well as their physiology, more accurate and detailed representation of different groups can be generated to improve these model predictions (Tagliabue, 2023; Cetinić et al., 2024). Knowledge of PCC may also further help estimate carbon export flux (Kramer et al., 2024a). Additionally, monitoring for harmful algal blooms (HABs) is important for human health as well as understanding their impacts on fisheries and ecosystems (Anderson et al., 2012), and fisheries managers may benefit from knowledge of PCC to aid stock assessments and identify suitable fishing

zones (Satterthwaite et al., 2023; Sathyendranath et al., 2023).

A common approach to estimate phytoplankton abundances and determine PCC is the measurement of phytoplankton pigments via high-performance liquid chromatography (HPLC). Specifically, certain accessory pigments can be used as proxies, or diagnostic pigments, to determine the abundances of specific groups, even though some of these pigments are shared among

groups (Jeffrey et al., 2011; Kramer and Siegel, 2019). As pigments directly alter the shapes and magnitudes of remote sensing reflectance spectra, HPLC-measured pigments are also extensively used to develop and validate satellite-based remote sensing algorithms. Given the recent availability of hyperspectral remote sensing reflectance data from NASA's Plankton, Aerosol, Cloud, ocean Ecology (PACE) mission, the continued development of remote sensing algorithms based on HPLC pigments may enable global scale estimation of phytoplankton pigment concentrations and PCC (Kramer et al., 2022; Cetinić et al.,

65 2024).



As a result, it is important to compare phytoplankton pigments to other metrics of PCC. If other metrics agree with pigments, they may also be useful for validating remote sensing algorithms. Conversely, it is important to compare other metrics of PCC to phytoplankton pigments as it remains uncertain whether indirect approaches such as DNA and RNA sequencing are

reflective of phytoplankton biomass. DNA metabarcoding is increasingly being used to determine PCC, primarily by sequencing hypervariable regions of the 16S gene for prokaryotes and phytoplankton plastids or the 18S gene for eukaryotes (Lopes dos Santos et al., 2022), and it is increasingly common to use environmental mRNA sequencing, or metatranscriptomics, to determine PCC (Cohen et al., 2022).

Initial comparisons between phytoplankton pigments and DNA indicate that the metrics are correlated in some cases; however, these comparisons have been hindered by the compositional nature of sequencing data (Catlett et al., 2023; Kramer et al., 2024b). By default, DNA and RNA sequencing produce relative abundance data, which complicates their interpretability and can lead to spurious correlations (Gloor et al., 2017). This is because there are multiple scenarios that could lead to a taxonomic group having the same or different relative abundances among samples, and potential differences in relative abundances are

not necessarily reflective of differences in their true abundances. For example, one group's true abundance could be the same in two samples while another group's is lower in just one sample leading to a higher relative abundance of the first group even though the two abundances are equal. Alternatively, both groups' true abundances could be higher in one sample but to different degrees, resulting in lower relative abundances in one group even though its true abundance has increased.

To compare phytoplankton pigments and relative DNA abundances, previous studies have also normalized diagnostic pigment concentrations to total chlorophyll $a$ concentrations or the sum of diagnostic pigments, thereby making both DNA and pigment data relative quantities (Catlett et al., 2023; Kramer et al., 2024b). However, the use of these approaches while only using relative abundances of prokaryotic or eukaryotic taxa presents other issues. As an example, high total chlorophyll $a$ could be driven by relatively high cyanobacterial abundances that would not be captured in 18S sequencing, which targets eukaryotes.

If the eukaryotic phytoplankton community is then dominated by one group, this group would have a high relative abundance that is coupled to low concentrations of their diagnostic pigment relative to total chlorophyll $a$. The use of 16S alone to capture both prokaryotic cyanobacteria and eukaryotic plastids may circumvent this issue, but references for plastid 16S sequences are not as comprehensive and are influenced by potentially greater variability in plastid copy number (Lopes dos Santos et al., 2022; Decelle et al., 2015). Alternatively, simultaneous metabarcoding of the 16S and 18S rRNA gene with three domain

primers addresses this issue, but it may also require greater sequencing depth to capture the diversity of the eukaryotic community (Yeh et al., 2021). Even if eukaryotic relative abundances are only compared to the sum of eukaryotic diagnostic pigments, the DNA-based relative abundances are still influenced by non-photosynthetic taxa. Altogether, these problems with normalization and potential discrepancies between true abundances and relative metrics can be severe issues that produce misleading correlations.




The estimation of absolute abundances via internal standards avoids these issues and enables comparisons between gene or transcript concentrations and phytoplankton pigment concentrations (Lin et al., 2019; Cohen et al., 2022). This approach was used in a study in the Western Antarctic Peninsula to compare 18S DNA concentrations to group-specific chlorophyll *a* concentrations (Lin et al., 2019). With a small sample size ($n$ = 16) and cryptophyte-dominated communities, strong

relationships for cryptophytes and *Phaeocystis* were observed, but not with diatoms. Other groups were not examined, and it remains unclear if these relationships persist with larger sample sizes, other regions with different communities, or across the entire spectrum of phytoplankton abundances. Furthermore, it remains unclear if absolute quantities of mRNA from metatranscriptomics and phytoplankton pigments relate to one another.

Here we investigate relationships between phytoplankton pigments and quantitative DNA- or RNA-based abundances with paired samples collected over a seven-year period on seasonal California Cooperative Oceanic Fisheries Investigations (CalCOFI) surveys. The CalCOFI sampling area is within the California Current Ecosystem, a coastal upwelling biome associated with an eastern boundary current. Although the sampling area is largely restricted to the southern California Current (Figs. 1A and 1B), it is along a major biogeographic boundary at Point Conception, and contains phytoplankton taxa commonly

associated with both the central North Pacific Subtropical Gyre and the subarctic Northeast Pacific; thus, it captures communities associated with the broader region (Checkley and Barth, 2009; Venrick, 1998). The range in phytoplankton abundances closely aligns with those observed globally and is captured by sampling the gradients from nearshore stations influenced by upwelling to offshore oligotrophic stations (Deutsch et al., 2021; James et al., 2022; Venrick, 2002). CalCOFI additionally conducts four cruises per year, capturing differences from seasonal and interannual variability. Beyond examining

PCC, we explore additional applications stemming from relationships between phytoplankton pigments and DNA- or RNA-based abundances including investigating productivity-diversity relationships, forecasting harmful algal blooms, and inferring expression of specific genes.





**Figure 1. Overview of CalCOFI samples used in this study. (A, B) Locations and quantity of samples for DNA and RNA with**
**corresponding phytoplankton pigment samples. (C, D) Temperature (°C) and nitrate concentrations (µmol L$^{-1}$) across seasons for**
**the paired DNA and pigment samples used here. Corresponding data for the RNA samples are shown in Fig. S1. (E) Phytoplankton**
**pigment concentrations. Pigment abbreviations: TChla, total chlorophyll *a*; Allo, alloxanthin; ButFuco, 19'-**
**butanoyloxyfucoxanthin; DVChla, divinyl chlorophyll *a*; Fuco, fucoxanthin; HexFuco; 19'-hexanoyloxyfucoxanthin; MVChlb,**
**monovinyl chlorophyll *b*; Perid, peridinin; Zea, zeaxanthin. (F, G) Relative abundances of different phytoplankton groups using the**
**18S-V4 (red), 18S-V9 (blue), or transcript abundance (metaT, green). For prokaryota, the 16S V4-V5 region was used and is shown**
**in red. (H, I) Absolute abundances for 18S or 16S rRNA genes (copies L$^{-1}$, left y-axis) and transcripts (transcripts L$^{-1}$, right y-axis).**
**Different metrics are colored as in panels F and G.**




## 2 Materials and Methods

### 2.1 Sample collection and biogeochemical measurements

From select stations on seasonal CalCOFI cruises from 2014 to 2020, 417 DNA and 118 RNA samples were collected concurrently with phytoplankton pigments within the euphotic zone along CalCOFI lines 80.0 and 90.0 and at the Santa Barbara Basin (Station 81.8 46.9) (Figs. 1A and 1B). These data represent only a subset of the on-going NOAA-CalCOFI Ocean Genomics (NCOG) time series, with no DNA samples from 2017 and only RNA samples from 2017-2020 to examine samples only where quantitative approaches for DNA and RNA were employed with concurrent sampling of phytoplankton

pigments (James et al., 2022). CalCOFI cruises survey the California Current Ecosystem in a standardized grid pattern with most stations in the southern California Current region. During this time period, winter cruises occurred during January and February, spring cruises occurred during April, summer cruises occurred in July and August, and autumn cruises occurred in October and November. There was no Spring cruise during 2020 due to the COVID-19 pandemic. At each station, seawater was collected from the near-surface (normally 10 m) and the subsurface chlorophyll maximum layer (SCML) with a CTD

rosette for pigments, DNA, RNA, and flow cytometry. Temperature data are derived from duplicate Seabird SBE 3Plus sensors on a Seabird 911+CTD. Macronutrient concentrations were measured on a QuAAtro continuous segmented flow autoanalyzer (SEAL Analytical) alongside reference materials (KANSO technos) (Armstrong et al., 1967; Gordon et al., 1992). The nitracline depth is defined as the depth where nitrate first exceeds 1 µM.

Primary productivity was estimated via $^{14}$C uptake at select stations. Briefly, seawater was collected from six depths representing 56%, 30%, 10%, 3%, 1%, and 0.3% surface light levels shortly before local apparent noon and dispensed into triplicate 250 mL polycarbonate bottles (two light bottles and one dark control). Bottles were inoculated with NaH$^{14}$CO$_3$ and incubated until civil twilight in tubes with flow-through seawater and neutral-density screens to simulate *in situ* light levels. Following incubation, samples were filtered onto 0.45 µm HA filters (Millipore), acidified with HCl, immersed in scintillation

fluor, and measured with a scintillation counter once back onshore. Half-day productivity at each depth was averaged between the two light bottles and corrected with the dark uptake bottle then multiplied by 1.8 to obtain 24 hour productivity (Eppley, 1992). When comparing productivity to diversity from DNA, samples from the entire NCOG dataset (2014 to 2020) that were closest to the collection depths of productivity samples were used (*n* = 757).

### 2.2 High-performance liquid chromatography pigment analysis

Phytoplankton pigment concentrations were determined with high-performance liquid chromatography (HPLC). Samples were collected with 0.5, 1.04, or 2.2L opaque brown bottles depending on the fluorescence measured by the CTD and filtered onto 25 mm GF/F filters under low vacuum pressure (≤ 40 mm Hg). Once completed filtering, the filters were carefully folded in half, blotted on a paper towel to remove excess water, and stored in 2 mL cryovials in liquid nitrogen until analysis at the Horn



Point Analytical Services Laboratory at the University of Maryland. Taxon-specific contributions to total chlorophyll *a*
concentrations were determined with phytoclass v1.0.0 (Hayward et al., 2023).

## 2.3 Nucleic acid sample collection and extraction

For DNA samples, 0.2 to 10.4 L (mean = 3.3 L) of seawater was filtered onto 0.22 µm Sterivex™ filters. RNA samples were
collected simultaneously following the same approach but with generally higher volumes (2.0 to 4.8 L, mean = 4.0 L). RNA
samples were also only collected near local apparent noon to minimize bias from the diel cycle.


Following filtration, samples were immediately flash frozen in liquid nitrogen and stored at -80℃. DNA was extracted with
the Macherey-Nagel NucleoMag Plant kit on an Eppendorf epMotion 5075TMX and assessed on a 1.8% agarose gel. At the
start of DNA extraction during the addition of lysis buffer, 1.74 to 3.78 ng of *Schizosaccharomyces pombe* genomic DNA was
added to each sample as an internal standard (Lin et al., 2019). RNA was also extracted on the Eppendorf epMotion but with
the Machery-Nagel NucleoMag RNA kit. As an internal standard, 2 to 5 billion copies of Invitrogen™ ArrayControl™ RNA
Spikes #1 and #8 were added to the lysis buffer of each sample at a 2.66:1 ratio.

## 2.4 Amplicon library preparation and sequencing

Amplicon libraries separately targeting the V4-V5 region of the 16S rRNA gene, the V4 and V9 regions of the 18S rRNA
gene, and the ITS2 gene from the diatom genus *Pseudo-nitzschia* were constructed via a one-step PCR with the TruFi DNA
Polymerase PCR kit to simultaneously amplify the region of interest and barcode each sample. For 16S, the 515F-Y (5′-GTG
YCA GCM GCC GCG GTA A-3′) and 926R (5′-CCG YCA ATT YMT TTR AGT TT-3′) primer set was used (Parada et al.,
2016). For 18S-V4, the V4F (5′-CCA GCA SCY GCG GTA ATT CC-3′) and V4RB (5′-CCA GCA SCY GCG GTA ATT
CC-3′) primer set was used (Berdjeb et al., 2018). For 18S-V9, the 1389F (5′-TTG TAC ACA CCG CCC-3′) and 1510R (5′-
CCT TCY GCA GGT TCA CCT AC-5′) primer set was used (Amaral-Zettler et al., 2009). For *Pseudo-nitzschia* ITS2, the
5.8SF (5′-TGC TTG TCT GAG TGT CTG TGG A-3′) and 28SR (5'-TAT GCT TAA ATT CAG CGG GT-3′) primer set was
used (Lim et al., 2018).

Each reaction was performed with an initial denaturing step at 95℃ for 1 minute followed by 30 cycles of 95℃ for 15 seconds,
56℃ for 15 seconds, and 72℃ for 30 seconds. 2.5 µL of each PCR reaction was run on a 1.8% agarose gel to confirm
amplification, then PCR products were purified with Beckman Coulter AMPure XP beads following the manufacturer's
instructions. PCR quantification was performed in duplicate using the Invitrogen Quant-iT PicoGreen dsDNA Assay kit.
Samples were then combined in equal proportions into multiple pools followed by another 0.8x AMPure XP bead purification
on the final pool. DNA quality of each pool was evaluated on an Agilent 2200 TapeStation, and quantification was performed
with the Qubit HS dsDNA kit. Each 16S or 18S pool was sequenced on an Illumina MiSeq (2 x 300 bp for 16S and V4 or 2 x
150 bp for V9) except for the one pool for the 2014-2016 euphotic zone V9 samples, which was run on an Illumina NextSeq



(Mid Output, 2 x 150 bp). The *Pseudo-nitzschia* ITS2 pool was sequenced on a single lane of an Illumina NovaSeq 6000 with a SP flow cell (2 x 250 bp).

**2.5 Analysis of amplicon sequence data**

Amplicons were first analyzed with QIIME2 v2019.10 for the 16S and 18S data or QIIME2 v2021.2 for the *Pseudo-nitzschia*
ITS2 data (Bolyen et al., 2019). Briefly, paired-end reads were trimmed to remove adapter and primer sequences with cutadapt (Martin, 2011). Trimmed reads were then denoised with DADA2 to produce amplicon sequence variants (ASVs). Each MiSeq run was denoised with DADA2 separately to account for different error profiles in each run then merged. Taxonomic annotation of ASVs was performed with the q2-feature-classifier naïve bayes classifier using the SILVA database (Release 138) for 16S ASVs and the PR$^2$ database (v4.13.0) for 18S ASVs (Bokulich et al., 2018; Pedregosa et al., 2011; Guillou et al.,
2012; Pruesse et al., 2007). For ITS2 ASVs, annotation was performed using BLAST (classify-consensus-blast) with *Pseudo-nitzschia* ITS2 sequences database as references (Brunson et al., 2024; Lim et al., 2018). When examining 16S relative abundances, all eukaryotic, plastid, and mitochondrial ASVs were removed, and when examining 18S relative abundances, dinoflagellates exclude Syndiniales and prymnesiophytes refers to all Prymnesiophyceae unless otherwise noted.

To estimate absolute abundances of 16S and 18S ASVs, recovery of the aforementioned internal standards was used (Lin et al., 2019). For each ASV within each sample, the number of reads was divided by the ratio of *T. thermophilus* or *S. pombe* reads to the number of rRNA copies added. The total number of copies was then normalized to the volume filtered for each sample to estimate copies L$^{-1}$.

**2.6 Metatranscriptome sequencing, assembly, and analysis**

To examine eukaryotic sequences, poly-A selected RNA-Seq libraries were created. Poly-A selected cDNA from total RNA was generated with the SMART-Seq® v4 Ultra® Low Input RNA Kit for Sequencing (Takara Bio USA, Inc.), which was then sheared with a Covaris® E210 focused-ultrasonicator targeting 300 bp fragments. The final sequencing library was then constructed with the NEB NEBNext® Ultra™ II DNA Library Kit and sequenced on three lanes of a NovaSeq 6000 with a S4 flow cell (2 x 150 bp).


For prokaryotic sequences, sequencing libraries were constructed following ribosomal RNA (rRNA) depletion. For samples from 2014-2019, rRNA was depleted with a 2:1:1 mixture of Ribo-Zero Plant, Bacteria, and Human/Mouse/Rat (Illumina) following the manufacturer's low input protocol. For samples from 2020, rRNA depletion was performed with siTOOLs riboPOOLs with a 6:1:1 mixture of the pan-prokaryote, pan-plant, and pan-mammal riboPOOLs following the manufacturer's
instructions for low inputs. rRNA depletion was confirmed with all samples on an Agilent TapeStation 2200 with High Sensitivity RNA ScreenTape. cDNA was then synthesized using the Ovation RNA-Seq System V2 kit (NuGEN) using both poly-A and random hexamer primers followed by fragmentation with a Covaris® E210 focused-ultrasonicator targeting 300



bp fragments. cDNA was then purified with Agencourt RNAClean XP beads, and library preparation was performed with the Ovation Ultralow System V2 (NuGEN). After end repair, ligation, and amplification, the libraries were assessed on an Agilent TapeStation 2200 with High Sensitivity DNA ScreenTape. A subset of samples from 2014-2016 were sequenced separately in four pools on an Illumina HiSeq 4000 (2 x 150 bp). The remaining libraries were sequenced across five pools on an Illumina NovaSeq 6000 with a S4 flow cell (2 x 150 bp).

For both types of libraries, the resulting raw reads were trimmed for quality and to remove Illumina adaptors. Ribosomal RNA sequences were also removed with Ribopicker v0.4.3 (Schmieder et al., 2011). Trimmed and filtered reads were then used for assembly into contigs, and abundances were quantified by mapping these reads to the assembly. Both assembly and read mapping were performed with CLC Bio Genomics Server v21.0.3. Gene prediction was performed with FragGeneScan v1.16, and rRNA removal was performed again with Ribopicker (Rho et al., 2010). Predicted proteins were further filtered to remove those less than 10 amino acids long or with greater than or equal to 20% stop codons. Gene clusters were generated from the predicted proteins with MCL with the inflation option (-I) set to 4 and scheme option (-scheme) set to 6 (Enright et al., 2002).

With the final assemblies, taxonomic annotation of each protein coding gene was assigned via DIAMOND BLASTP alignments using PhyloDB v1.076 as a reference database (Bertrand et al., 2015). Final taxonomic assignments were based on highest Lineage Probability Index values (Podell and Gaasterland, 2007). For functional annotation, DIAMOND BLASTP alignments were performed with the Kyoto Encyclopedia of Genes and Genomes (KEGG; Release 94.1) (Kanehisa et al., 2017). KEGG Ortholog (KO) assignment was performed with KofamKOALA, which uses hmmsearch against KOfam, an HMM database of KOs (Aramaki et al., 2019). E-value cutoffs of 1e-5 were used throughout.

To normalize read counts, raw reads were first converted to transcripts per million (TPM) (Li et al., 2009). Absolute quantities (transcripts $L^{-1}$), i.e. quantitative metatranscriptomics, were then calculated based on the average recovery of the aforementioned Invitrogen™ ArrayControl™ RNA Spikes and the volume of sample filtered as described in Cohen et al. (2022). For performing correlations with individual genes, transcripts $L^{-1}$ within each taxonomic group and sample were summed by KO annotation, and if KO annotation was absent, the KEGG gene annotation was used. The domoic acid biosynthesis gene, *dabA*, was annotated using the sequences reported in Brunson et al. (2018).

**2.7 Flow Cytometry**

Two mL samples of seawater were collected in cryovials and fixed with 100 µL of 0.2 µm filtered 10% paraformaldehyde for a 0.5% final concentration. Fixation was allowed to occur for 10 min before freezing the sample in liquid nitrogen. Once onshore, samples were stored at -80°C or on dry ice until analysis at the SOEST Flow Cytometry Facility, University of Hawaii at Manoa.





Prior to analysis, thawed samples were stained with Hoechst 33342 (1 µg mL$^{-1}$, v/v, final concentration) at room temperature in the dark for 1 hour (Monger and Landry, 1993). Aliquots (100 µl) were analyzed using a Beckman-Coulter EPICS Altra flow cytometer with a Harvard Apparatus syringe pump for volumetric sample delivery. Simultaneous (co-linear) excitation of the plankton was provided by two argon ion lasers, tuned to 488 nm (1 W) and the UV range (200 mW). The optical filter

configuration distinguished populations based on chlorophyll $a$ (red fluorescence, 680 nm), phycoerythrin (orange fluorescence, 575 nm), DNA (blue fluorescence, 450 nm), and forward and 90° side-scatter signatures. Blue-fluorescence and red-fluorescence signals are used to distinguish DNA-containing heterotrophic (non-pigmented) from phototrophic (chlorophyll-containing) cells. Standardized fluorescence and scatter parameters were determined relative to 0.5- and 1.0-µm yellow-green calibration beads and 0.5-µm UV calibration beads run in each sample. Bead-normalized red-fluorescence values

are also used as a measure of cellular chlorophyll $a$ (Landry et al., 2003). Raw data (listmode files) were processed using the software FlowJo (Treestar Inc., www.flowjo.com). Cell abundance estimates were then calculated accounting for the volume dilutions from the added preservative and stain solutions.

### 2.7 Statistics

All correlations and models were generated with R v4.3.2. Generalized additive models (GAMs) were fit with the mgcv

package v 1.9-1 (Wood, 2017). When examining multiple gene correlations, $P$-values were adjusted using the Benjamini & Hochberg procedure (Benjamini and Hochberg, 1995).

## 3 Results and Discussion

### 3.1 Quantitative relationships among pigments, DNA, and RNA

As the sampling locations span nearshore stations influenced by coastal upwelling to offshore oligotrophic conditions (James

et al., 2022; Venrick, 2002), the paired phytoplankton pigment and DNA or RNA samples encompass a wide range of environmental conditions (Figs. 1A-1D and S1). Seasonal and interannual variability were also captured with notably lower temperatures and higher nutrients in the Spring season relating to seasonal upwelling as well as an unprecedented marine heatwave from mid-2014 to 2016 associated with El Niño in 2015 (Di Lorenzo and Mantua, 2016; Jacox et al., 2018). Concurrent with the wide-ranging environmental conditions, total chlorophyll $a$ and accessory pigment concentrations often

spanned several orders of magnitude with all accessory pigments detected in concentrations as low as 0.001 to 0.005 µg L$^{-1}$ (Fig. 1E). Fucoxanthin (Fuco) exhibited the greatest range with concentrations reaching 6.81 µg L$^{-1}$, and 19'-hexanoyloxyfucoxanthin (HexFuco) exceeded 1 µg L$^{-1}$ in several samples. All other pigment concentrations examined here were always less than 1 µg L$^{-1}$.

DNA metabarcoding of both the V4 and V9 regions of the 18S rRNA gene for eukaryotes as well as the V4-V5 regions of the 16S rRNA gene for prokaryotes similarly showed wide ranges in relative abundances (Figs. 1F and 1G). Among eukaryotes,



diatoms exhibited the greatest range in accordance with their known dominance during blooms in the region (Goericke, 2011; Venrick, 2012). Chlorophytes also exhibited a wide range of relative abundances, and their average abundance exceeded that of diatoms. Dinoflagellates were also found in high abundances, and while they indeed bloom in the region (Kahru et al., 2021;

Anderson et al., 2008), their high abundances via 18S metabarcoding may be due to bias from high 18S gene copy numbers (Martin et al., 2022). Other eukaryotic phytoplankton groups were typically less than 10% of the total eukaryotic community. In particular, cryptophytes were less than 1% of the community on average but were found to be as high as 5%. Within eukaryotic mRNA expressed as relative abundances [transcripts per million (TPM)], intragroup variability was less pronounced, and most groups displayed higher relative abundances compared to their 18S DNA abundances (Figs. 1F). The

sole exception was with dinoflagellates which had lower relative abundances with RNA compared to DNA, further suggesting bias in their DNA relative abundances due to high 18S DNA copy numbers.

Within prokaryotes, all cyanobacteria combined and the cyanobacterial genus *Prochlorococcus* displayed similar distributions, indicating that *Prochlorococcus* was often the dominant member of the cyanobacterial community (Figs. 1G and 1I). Cell

abundances measured via flow cytometry also show that *Prochlorococcus* is dominant in these samples, with *Prochlorococcus* displaying higher abundances than *Synechococcus* in 76% of samples (Fig. S2). Cyanobacteria as a whole and *Prochlorococcus* also displayed lower relative abundances from transcripts compared to 16S rRNA genes (Fig. 1G).

When converted to absolute abundances based on the recovery of internal standards and normalizing to the volume filtered

(gene copies L$^{-1}$ or transcripts L$^{-1}$), differences and distributions among groups were similar to those observed with relative abundances (Figs. 1F-1I). However, such quantitative approaches have not been widely adopted, and most metabarcoding data are solely expressed as relative abundances. As previously described, other studies have normalized phytoplankton pigments to total chlorophyll *a* concentrations in an attempt to make DNA-based relative abundances and phytoplankton pigments comparable (Catlett et al., 2023; Kramer et al., 2024b). When using pigments directly, diagnostic pigments used for specific

phytoplankton groups include monovinyl chlorophyll *b* (MVChlb) for chlorophytes, alloxanthin (Allo) for cryptophytes, fucoxanthin (Fuco) for diatoms, peridinin (Perid) for dinoflagellates, 19'-butanoyloxyfucoxanthin (ButFuco) for pelagophytes, 19'-hexanoyloxyfucoxanthin (HexFuco) for prymnesiophytes, zeaxanthin (Zea) for cyanobacteria, and divinyl chlorophyll *a* (DVChla) for *Prochlorococcus* (Catlett et al., 2023; Kramer and Siegel, 2019).

Within eukaryotic taxa, comparisons between normalized pigment concentrations and DNA-based relative abundances showed moderate to strong correlations for several groups (Fig. 2A). In particular, the relative abundances of diatoms and Fuco displayed the strongest agreement ($r$ = 0.79-0.82) followed by pelagophytes and ButFuco ($r$ = 0.71-0.77). Cryptophytes with Allo and chlorophytes with MVChlb were also moderately well correlated ($r$ = 0.51-0.65) whereas dinoflagellates with Perid and prymnesiophytes with HexFuco were weakly correlated ($r$ = 0.32-0.42). Previous studies using samples from the California

Current (Plumes and Blooms), North Atlantic Ocean (NAAMES), and Northeast Pacific Ocean (EXPORTS) have shown



similar results where relationships are notable for chlorophytes, cryptophytes and pelagophytes, weak for prymnesiophytes and dinoflagellates, and inconsistent among the studies for diatoms (Catlett et al., 2023; Kramer et al., 2024b). In another previous study in the Western Antarctic Peninsula where a chemotaxonomic approach that partitions total chlorophyll *a* (CHEMTAX) was used rather than diagnostic pigments directly, cryptophytes were moderately well correlated between the

measurements whereas diatoms were not (Lin et al., 2019; Mackey et al., 1996). Within these studies, abundances for some groups were also too low to be included.

Although these results suggest that there are moderately strong relationships between pigments and DNA-based relative abundances for several phytoplankton groups, these analyses are hindered by the compositional nature of relative abundance

data as previously described. The estimation of absolute abundances via internal standards removes compositionality and enables the comparison of DNA- or RNA-based concentrations (copies $L^{-1}$ or transcripts $L^{-1}$) to phytoplankton pigment concentrations ($\mu g\ L^{-1}$). By doing so here, all correlations were stronger than those with relative abundances for eukaryotic phytoplankton taxa (Fig. 2B). In particular, the relationships between dinoflagellates and Perid was dramatically stronger with both marker genes ($r = 0.77$-$0.79$). The relationship for prymnesiophytes was also stronger but not to the same degree ($r =$

$0.46$-$0.51$). Furthermore, all groups were most strongly correlated with their diagnostic pigments except prymnesiophytes, which had similarly strong correlations with peridinin (Fig. S3). By using taxon-specific biomass as chlorophyll *a* concentrations with a chemotaxonomic approach (phytoclass) (Hayward et al., 2023), correlations were often similar (Fig. S4). Moreover, strong correlations were also observed among absolute transcript abundances and pigment concentrations ($r = 0.66$-$0.88$) except for prymnesiophytes which had a similar correlation when using their DNA-based abundances ($r = 0.43$) (Fig.

2C).

These results indicate that abundances of diagnostic phytoplankton pigments and phytoplankton DNA or RNA often agree. Furthermore, they suggest that the aforementioned issues with compositionality or normalization of pigments to total chlorophyll *a* may lead to misleading conclusions about these relationships, particularly for dinoflagellates. Besides

compositionality, DNA- and RNA-based abundances have other biases that can also lead to discrepancies. Copies of 18S DNA can vary orders of magnitude among closely related phytoplankton species (Zhu et al., 2005), and DNA sequencing may also capture deceased cells that have lost some or all of their pigments. RNA abundances may be variable within a cell, instead reflecting overall activity or differential transcription rates in response to environmental stimuli (Moran et al., 2012). Both metrics are also influenced by the comprehensiveness of reference databases where unknown phytoplankton sequences may

be unassigned (Krinos et al., 2023).




**Figure 2. Correlations among abundance metrics for each major eukaryotic phytoplankton group.** Pearson correlations are all significant (*P* < 0.05) and coefficients are displayed in each panel. Lines with 95% confidence intervals show linear models with significant relationships between variables (*P* < 0.05). (A) Diagnostic pigments normalized to total chlorophyll *a* (TChla) against relative abundances of either the 18S-V4 (red) and 18S-V9 (blue) rRNA gene abundance. (B) Diagnostic pigment concentrations against absolute abundances of the 18S-V4 (red) and 18S-V9 (blue) rRNA genes (copies L⁻¹). (C) Diagnostic pigment concentrations against absolute abundances of mRNA (transcripts L⁻¹).



Differences in pigment abundances may also be a source of variability. For example, reduced light availability may lead to
cellular increases in accessory pigments (Henriksen et al., 2002). Using mixed layer depth as a proxy for light history,
decreased light availability has been shown to have a small but significant effect on discrepancies between phytoplankton
pigments and DNA relative abundances (Catlett et al., 2023). To examine the effects of light history, we performed linear
regressions on the residuals from linear models for absolute abundances with depth as a predictor variable and coarse proxy
for reduced light availability (Fig. S5). Besides with cryptophytes, depth was found to positively and significantly predict
residuals, aligning with an increase in pigments at depth which may be attributed to reduced light. High variability in near-
surface samples was also observed and may reflect large differences in light conditions from sampling throughout the diel
cycle.

Furthermore, as some pigments are shared among certain groups, the presence of one group may influence pigment
concentrations that are diagnostic of a different group. Fucoxanthin, for example, is not only found in diatoms but also other
photosynthetic eukaryotes with red algal-derived plastids (Jeffrey et al., 2011; Kramer and Siegel, 2019). Previous studies
suggest that shared pigments are an important source of disagreement between the two measurements, but the strong
correlations for many groups, particularly with diatoms, suggests that it may be a less prevalent issue in certain cases (Catlett
et al., 2023; Kramer et al., 2024b).

Intragroup variability in pigment composition may also be a contributing factor with different species having different pigment
concentrations (Zapata et al., 2004; Neeley et al., 2022). In particular, dinoflagellates may be influenced by taxa that no longer
have peridinin. The previously shown dinoflagellate correlations were performed without Syndiniales, an early branching clade
that are likely parasitic and have lost their plastid altogether (Decelle et al., 2022; Guillou et al., 2008). Overall, Syndiniales
accounted for 30 to 34% of dinoflagellate 18S copies in our data, and their removal considerably increased the strength of
dinoflagellate-peridinin correlations (Fig. S6 and Table S1). Other dinoflagellate taxa appear to have instead replaced their
plastids with others via kleptoplasty, integration of a new plastid, or obtaining endosymbionts (Novák Vanclová and Dorrell,
2024). For example, the Kareniaceae family (e.g., the genera *Karenia, Karlodinium*, and *Takayama*) and Kryptoperidiniceae
family (e.g., the genus *Durinskia*) contain fucoxanthin from an ancestral haptophyte or diatom endosymbiont respectively,
members of the genus *Dinophysis* have a complex kleptoplastidic relationship with a cryptophyte, and the genus *Lepidodinium*
contains plastids originating from a chlorophyte (Novák Vanclová and Dorrell, 2024; Kamikawa et al., 2015; Zapata et al.,
2012). However, neither Kryptoperidiniceae nor *Lepidodinium* were detected in these data with either 18S marker, and filtering
of other groups did not alter the strength of the correlation between Dinoflagellates and peridinin indicating that besides
Syndiniales, these groups do not influence the dinoflagellate-peridinin relationship in this region (Table S1). The
prymnesiophyte *Phaeocystis globosa* has also been suggested to contain ButFuco rather than HexFuco (Wang et al., 2022;
Antajan et al., 2004), and while filtering ASVs annotated as this species marginally improved the strength of the correlation
for prymnesiophytes with the V4 data (from $r = 0.46$ to $r = 0.48$), it was lower in the V9 data (from $r = 0.51$ to $r = 0.45$).





**Figure 3. Correlations among abundance metrics for cyanobacteria. (A)** Average bead-normalized red fluorescence per cell for *Prochlorococcus* versus depth (m). The fluorescence units are arbitrary. **(B)** *Prochlorococcus* DNA-based absolute abundances (copies L⁻¹) against cellular abundances. **(C, D)** Relative and absolute abundance comparisons between all cyanobacteria and zeaxanthin (Zea). **(E, F)** Relative and absolute abundance comparisons between *Prochlorococcus* and divinyl chlorophyll *a* (DVChla). Pearson correlations are significant ($P < 0.05$) and coefficients are displayed in panels B-F. Lines with 95% confidence intervals shows linear models with significant relationships between variables ($P < 0.05$).



Within cyanobacteria, *Prochlorococcus* chlorophyll content significantly increased with depth, suggesting an increase in pigments as light availability is reduced (Fig. 3A). Furthermore, *Prochlorococcus* absolute abundances strongly agreed with cell abundances from flow cytometry throughout the water column as previously observed in the greater region (Fig. 3B) (Jones-Kellett et al., 2024). In combination, these results suggest that DNA-based abundances and pigments are increasingly
uncoupled as light availability is reduced. To minimize effects from light, only samples from the upper 20 m of the water column were further examined. When using relative abundances, all cyanobacteria with Zea and *Prochloroccocus* with DVChla were strongly correlated (Fig. 3C-F). However, DVChla, was not detected in 12.5% of samples which generally corresponded to lower DNA abundances (Fig. 3F). Unlike the eukaryotic data, the strengths of these correlations were lower when using absolute abundances, albeit still moderately well correlated. Although variability in pigments was reduced by
removing samples deeper than 20 m, other factors such as cloud cover may further contribute the remaining observed pigment variability and a weaker correlation with absolute abundances. This variation in pigments may be normalized when dividing by total chlorophyll *a* leading to higher correlations with relative abundances. Some of the discrepancy with absolute abundances may also be caused by *Prochlorococcus* cell sizes which are smaller than the pore size of the GF/F filters used for pigments here (nominally 0.7 µm), leading to some *Prochlorococcus* cells being missed while being captured by flow
cytometry or DNA that used a smaller pore size (0.2 µm) (Ting et al., 2007; Partensky et al., 1999). As in eukaryotic phytoplankton, the absolute abundances of transcripts were also moderately well correlated for cyanobacteria with Zea and *Prochloroccocus* with DVChla (Fig. S7).

## 3.2 Applications from integrating phytoplankton pigments with molecular data

The strong agreement among eukaryotic phytoplankton pigments and DNA or RNA for several groups indicates that these
metrics are comparable proxies for phytoplankton abundances and community composition. As phytoplankton pigments directly impact remote sensing reflectance spectra, HPLC pigments are useful for validating remote-sensing algorithms; however, these results suggest that the absolute abundances of DNA or RNA may also be useful for model development. Furthermore, these results support that potential models for phytoplankton pigment concentrations via remote sensing may be able to provide comparable global estimates of different phytoplankton groups (Kramer et al., 2022). In the following sections,
we further examine relationships gleaned by integrating phytoplankton pigments and DNA- or RNA-based metrics to demonstrate potential applications for addressing ecological questions, monitoring harmful algal blooms, or inferring phytoplankton group-specific activity.

### 3.2.1 Increased taxonomic resolution with biomass estimation in ecological assessments

Phytoplankton pigments and DNA-based metabarcoding have separate strengths that can be leveraged when integrated:
phytoplankton contributions to total chlorophyll *a* can be estimated with chemotaxonomic approaches providing a standardized estimate of biomass for each group, and DNA offers marker gene-level resolution into the composition of each sample. This increased resolution is possible even when the DNA-based data are expressed solely as relative abundances.





To illustrate this combined approach, diversity expressed as the Shannon index (H') was compared to both total chlorophyll *a*
and taxon-specific chlorophyll *a* as proxies for phytoplankton biomass (Fig. 4). Biomass is often used as a proxy for productivity, thus enabling investigation of productivity-diversity relationships (PDRs) (Smith, 2007; Irigoien et al., 2004). PDRs may exhibit different trends, but marine phytoplankton are presumed to exhibit a unimodal distribution with maximum diversity at an intermediate level of productivity, including within models of phytoplankton communities in the California Current Ecosystem (Irigoien et al., 2004; Li, 2002; Goebel et al., 2013). PDRs may also be positive, negative, or flat, with
other studies suggesting that there is no relationship when accounting for potentially inadequate sampling (Cermeño et al., 2013; Smith, 2007). Examination of these relationships is particularly important for understanding how environmental change may impact diversity and productivity.

When comparing total chlorophyll *a* concentrations and the diversity of all six eukaryotic phytoplankton groups examined
here, the PDR for the region was negative rather than unimodal, with diversity remaining high at low and intermediate biomass levels before declining at high biomass (Fig. 4A). During the same cruises, additional samples concurrently measured DNA abundances and net primary productivity (NPP, mg C m$^{-3}$ d$^{-1}$). When comparing diversity and NPP directly, the relationship was similarly flat then negative (Fig. S8A). The Shannon index considers both richness and evenness, thereby downweighing the influence of rare taxa in comparison to richness, defined here as the number of ASVs (Ibarbalz et al., 2019; Ma, 2018).
Despite substantially more variability, richness also displayed a negative trend including when adding cyanobacterial richness (Fig. S8B and S9).

By partitioning total chlorophyll *a* concentrations into separate eukaryotic phytoplankton groups and leveraging the high taxonomic resolution of the DNA-based data, PDRs were examined within each group (Fig. 4). Dinoflagellates, diatoms, and
cryptophytes displayed significant relationships whereas other groups did not for both amplicons. Furthermore, diatoms and dinoflagellates displayed opposing trends, where diatoms displayed the expected unimodal relationship while dinoflagellates displayed a negative relationship. When considering richness, the trend for diatoms was largely positive, indicating a decline in evenness at the highest levels of biomass leading to lower diversity, and for cryptophytes, the relationship was slightly positive with both metrics (Fig. 4 and S10). Dinoflagellates had the highest richness with as many as 353 ASVs, while diatoms
had a maximum of 93 ASVs (Fig. S10). Cryptophytes and pelagophytes had low richness with maxima of 15 and 9 ASVs, respectively (Fig. S10). For pelagophytes, the relatively low richness and lack of trend may be due to dominance of a single species, *Pelagomonas calceolata,* that is highly prevalent in SCMLs (Guérin et al., 2022; Dupont et al., 2015). Meanwhile, chlorophytes and prymnesiophytes did not display significant relationships between biomass and richness. Considering the relatively high richness and sole negative trend, dinoflagellates appear to be the primary drivers of the community-wide trend,
besides at high biomass where diatoms have a high influence. This dominance by dinoflagellates is further evidenced by a shift to a more unimodal distribution by examining the community in the absence of dinoflagellates (Fig. S11).



**Figure 4. Productivity-diversity relationships in the region with biomass (chlorophyll *a* concentrations) as a proxy for productivity and diversity expressed as the Shannon Index (H') for both the 18S-V4 (red) and 18S-V9 (blue) data. (A) The diversity of all eukaryotic phytoplankton groups and total chlorophyll *a* concentrations. (B-D) Environmental variables against total chlorophyll *a* concentrations colored by near-surface (purple) or subsurface chlorophyll max (SCM, green) samples. (E-J) Diversity of individual phytoplankton groups against their taxon-specific biomass estimated with phytoclass. Lines represent GAMs and corresponding 95% confidence intervals where significant (*P* < 0.05). The deviance explained by each GAM or "NS" for not significant (*P* > 0.05) is shown above each panel for each amplicon.**

The observations of unimodal PDRs have led to hypotheses for the mechanisms that underlying them. Indeed, our observed negative trend aligns with the negative side of unimodal PDRs where diversity decreases with increased productivity. This decline is predicted to be associated with high productivity nearshore upwelling conditions where there is strong competition for light and opportunist large cells such as diatoms escape grazing (Goebel et al., 2013; Irigoien et al., 2004; Vallina et al., 2014). In agreement, the observed low diversity aligns with the nearshore environment, shallow nitracline depths, and increase in diatom biomass relative to dinoflagellates (Figs. 4 and S12). However, increased diatom richness with biomass indicates that the negative trend is not entirely driven by the dominance of a few opportunist diatom taxa that escape predation (Fig.



S10C). Rather, many diatoms appear to flourish when competition for nutrients is minimized under upwelling conditions, although evenness likely declines at the highest levels.


High diversity at intermediate productivity has been suggested to be associated with offshore oligotrophic conditions (Goebel et al., 2013). Although offshore oligotrophic samples aligned with the lowest biomass levels, they were in part responsible for intermediate NPP along with SCMLs at intermediate depths, in agreement with model predictions (Fig. 4 and S8). The low productivity and low diversity end of the unimodal distribution is expected to be caused by light limitation (Goebel et al.,

2013), and while chlorophyll *a* concentrations suggest that the deepest SCML samples align with intermediate biomass, they are influenced by reduced light causing elevated chlorophyll content (Cullen, 2015). As predicted, the deepest SCML samples displayed the lowest NPP rates; however, contrary to predictions, diversity and richness remained high in these samples resulting in an absence of the positive side of a unimodal distribution (Fig. S8).

The low productivity and diversity end of unimodal distributions have also been attributed to selective grazing with the dominance of a few slow-growing nutrient specialists (Vallina et al., 2014). As diversity and richness instead remained high, many phytoplankton taxa, particularly dinoflagellates, appear to coexist within low productivity regimes. Within dinoflagellates, this coexistence may be supported by mixotrophy or diel vertical migrations where nutrient availability at depth is exploited at night and photosynthesis in the near-surface occurs during the day (Zheng et al., 2023; Stoecker et al.,

2017). Within prymnesiophytes, there may also be some mixotrophic taxa (Koppelle et al., 2022). Small cells such as chlorophytes, pelagophytes, and prymnesiophytes are also at an advantage under oligotrophic conditions due to more effective resource acquisition and use (Raven, 1998), although their diversity is maintained across their biomass ranges (Fig. 4). Even though diatom diversity is lower under oligotrophic conditions, certain diatoms such as those that form symbiotic relationships with diazotrophs may be favored under these conditions and contribute to the increased diversity (Kemp and Villareal, 2018).


Overall, these results highlight the contrasting strategies of different phytoplankton groups and align with the classical view of diatoms and dinoflagellates on opposite ends of the r-selected vs K-selected continuum (Margalef, 1978). However, these results may differ from other regions that exhibit greater stability or experience frequent blooms of groups besides diatoms. Continued warming in the region is anticipated to lead to increased stratification resulting in conditions analogous to those in

the offshore oligotrophic region with deeper nitraclines (Ducklow et al., 2022; Lund, 2024). These projections imply that increased stratification leads to low productivity but high phytoplankton community diversity, although diatom diversity will be lower. In such a scenario, the high level of diversity will instead be driven by dinoflagellates and supplemented by a variety of picoeukaryotes, contrasting with predictions of lower diversity in ecosystem models of open ocean regions (Henson et al., 2021).





### 3.2.2 Monitoring and forecasting harmful algal blooms

The strong relationships between pigments and abundances of certain groups may also be useful for monitoring or forecasting harmful algal blooms (HABs). For example, certain species in the diatom genus, *Pseudo-nitzschia*, produce the neurotoxin domoic acid (DA) resulting in HABs worldwide (Bates et al., 2018). They are also commonly responsible for HABs in our study region and California Current at large (Lewitus et al., 2012). With absolute abundances accounting for 10% to 11% of diatom 18S copies, *Pseudo-nitzschia* was among the most dominant diatom genera, only exceeded by *Thalassiosira* and *Chaetoceros* (Fig. 5A). *Pseudo-nitzschia* was also detected in 74% to 79% of samples when diatoms were present with an equivalent number when fucoxanthin was detected. Overall, a strong positive relationship between *Pseudo-nitzschia* and total diatom abundances was observed, and fucoxanthin concentrations explained 10% more of the variance in *Pseudo-nitzschia* abundances than chlorophyll *a* concentrations (Fig. 5). Expression of *dabA*, the first gene in the domoic acid biosynthetic pathway (Brunson et al., 2018), was also detected in 13 metatranscriptomics samples, and 7 of the 11 samples with greater than 0.5 µg L$^{-1}$ concentrations of fucoxanthin had detectable *dabA* expression (Fig. 5E). These samples comprised diverse *Pseudo-nitzschia* species with an overall dominance of *P. delicatissima*; however, the highest *dabA* expression occurred when *P. australis* relative abundances were elevated (Fig. 5F).

Not all species of *Pseudo-nitzschia* have been shown to produce domoic acid, and DA production is influenced by environmental conditions (Bates et al., 2018). Similarly, expression of *dabA* does not always confer detectable particulate DA (Brunson et al., 2024). However, current models in the region that predict *Pseudo-nitzschia* HABs and domoic acid production use a variety of data including remotely-sensed chlorophyll *a* and two reflectance wavebands, suggesting that fucoxanthin detection in conjunction with other measurements and cellular modeling may offer better predictions for *Pseudo-nitzschia* and domoic acid (Anderson et al., 2016; Moreno et al., 2022).

Certain dinoflagellates may also cause HABs globally and in the region (Anderson et al., 2012; Anderson et al., 2021). These HABs are caused by certain species in the genera *Alexandrium, Dinophysis*, and *Gonyaulax* as well as the species *Gymnodinium catenatum* and *Lingulodinium polyedra* (Anderson et al., 2021; Trainer et al., 2010; Ternon et al., 2023). These genera were also among the most dominant dinoflagellate genera detected, although 39% of V4 and 55% of V9 18S copies for dinoflagellates were unassigned on a genus level (Fig. S13).





**Figure 5.** *Pseudo-nitzschia* abundances. **(A)** Relative abundances of the top three diatom genera based on total copies L⁻¹. **(B)** *Pseudo-nitzschia* absolute abundances against diatom (Bacillariophyta) absolute abundances. **(C)** *Pseudo-nitzschia* absolute abundances against total chlorophyll *a* concentrations **(D)** *Pseudo-nitzschia* absolute abundances against fucoxanthin concentrations. **(E)** Total *dabA* expression against fucoxanthin concentrations. **(F)** Relative abundances of *Pseudo-nitzschia* species from ITS2 sequencing (left y-axis) and total *dabA* expression (right y-axis). Samples are ordered by fucoxanthin concentrations as shown in Panel E.

Despite dinoflagellate diversity declining with increasing biomass, dinoflagellate richness was the highest among groups, including where the highest dinoflagellate biomass was observed (Fig. 4 and S10). As a result, the samples with high dinoflagellate abundances still comprise many genera. Considering that peridinin concentrations did not exceed 1 µg L⁻¹, an intense dinoflagellate bloom that may have resulted in even lower diversity was not captured here. Nevertheless, peridinin concentrations and V4-based abundances of the dinoflagellate genera *Alexandrium*, *Gonyaulax*, and *Gymnodinium* were





significantly related (Fig. S13). Peridinin concentrations also explained 9-16% more of the variance in the abundances of these genera compared to total chlorophyll *a* concentrations. In contrast, no significant relationships were observed for *Lingulodinium* and *Dinophysis,* the latter of which does not contain peridinin as previously noted. As in *Pseudo-nitzschia*, there are non-toxic members of these genera, and their presence does not imply the production of their respective toxins

(Anderson et al., 2012). However, the ability to distinguish increased peridinin or fucoxanthin concentrations with remote sensing suggests increases in the genera identified here and may aid HAB monitoring efforts.

### 3.2.3 Towards increased inference of biogeochemical or metabolic activity

Current ESMs are unable to confidently predict climate-driven changes to NPP, and improving these models to better account for phytoplankton abundances and functions is a critical component to address this uncertainty (Kwiatkowski et al., 2020;

Tagliabue, 2023). Model parameters include biological rates and biogeochemical fluxes, and there is increasing interest in the ability to connect 'omics data with rates such that the 'omics can inform these model parameters (Strzepek et al., 2022; Saito et al., 2024). The absolute quantities of certain proteins have shown promise for inferring rates of nitrite oxidation and carbon fixation (Saito et al., 2020; Roberts et al., 2024), although it is unclear if absolute transcript abundances will be able to serve a similar purpose (McCain et al., 2024).


The strong correlations between pigments and total transcript abundances in most groups examined here suggests that pigment abundances may also relate to the expression of specific genes (Fig. 2C). Within each group, the absolute abundances of genes clustered by KEGG annotations (Sect. 2.6) were correlated with their respective diagnostic pigments (Fig. 6). In chlorophytes, there was only one strongly significant correlation with a light-harvesting chlorophyll-binding protein (LHCB) and

*Prochlorococcus* which had no significant correlation. In other groups however, 67 to 2,312 genes were strongly correlated ($r > 0.60$, FDR $< 0.05$). For some taxa, specific genes displayed stronger correlations than both 18S rDNA and all transcripts combined (Fig. 2), indicating that they may individually be used as indicators of abundances. These include the pentose phosphate pathway genes 6-phosphogluconate dehydrogenase (*PGD*) and glucose-6-phosphate dehydrogenase (*G6PD*) in dinoflagellates as well as several accessory light harvesting complex (*LHCF*) proteins in prymnesiophytes. With 34% to 65%

of genes with no assigned function, the remaining genes fall into diverse sets of metabolic categories (Fig. 6).

In cryptophytes and dinoflagellates, several genes related to carbon fixation and photosynthesis were strongly correlated (Fig. 6). Cryptophytes and diatoms also showed strong correlations with several nitrogen metabolism genes. In baker's yeast (*Saccharomyces cerevisiae*), the abundances of individual proteins are generally poor predictors of their corresponding

reaction rates (McCain et al., 2024), but the combined abundances of functional units of genes, i.e. modules or subsystems, that are responsible for specific pathways may more accurately predict rates. Similarly, increases in the abundances of specific pigments here is indicative of the expression of certain pathways which may be useful in predicting group-specific reaction



rates *in situ*. Further establishment of these relationships or between pigments and protein abundances may potentially support global estimates of these group-specific reaction rates.




Figure 6. Strongly significant correlations between diagnostic pigments and gene expression (transcripts L$^{-1}$) associated with each respective group *(r > 0.60, FDR < 0.05)* and organized by KEGG module classes. To examine correlations, genes were aggregated 600 by KEGG Ortholog annotations and if not present, the KEGG gene annotations were used.



## 4 Conclusions

By integrating phytoplankton pigments with quantitative abundances of 18S DNA and total mRNA via metabarcoding and metatranscriptomics respectively, we demonstrate that diagnostic pigments for specific eukaryotic phytoplankton groups
correlate with both their DNA- and RNA-based abundances. Although there are inherent biases associated with each of these measurements, their relationships suggest that they are comparable and may be useful for the development of satellite-based remote sensing models of phytoplankton group-specific abundances. Rather than the 18S rRNA gene which suffers from variable copy numbers, other sequencing-based markers may be more useful for drawing these comparisons. For example, the photosystem gene *psbO* is universal among phytoplankton which normally have one or two at most copies per genome (Pierella
Karlusich et al., 2023). It is also unclear if 18S sequencing of rRNA rather than the rRNA gene from DNA offers a less biased assessment. However, these results also provide increased confidence that both DNA- and RNA-based abundances are reflective of phytoplankton group-specific biomass. Prymnesiophytes displayed the weakest correlations out of all groups examined here, but the calcium carbonate shells of coccolithophores are highly optically refractive enabling easier detection via satellite-based remote sensing (Balch, 2018). When coupled to other satellite, glider, or float-based measurements of
photophysiology (Lin et al., 2016; Ryan-Keogh et al., 2023), phytoplankton group-specific abundances, community composition, and physiological assessments may be able to be remotely and collectively assembled.

Although this assessment was only performed in the California Current Ecosystem, these relationships may extend to other regions. Previous use of quantitative metabarcoding in the Western Antarctic Peninsula with relatively high abundances of
cryptophytes showed that their chemotaxonomic abundances were well correlated with their DNA-based abundances (Lin et al., 2019). Diatoms were not strongly correlated, but this result may be an artifact from a small sample size that did not capture a large range of diatom abundances in the region. Considering that the strengths of correlations in all eukaryotic groups improved when translating relative abundances into absolute abundances here, previous studies in the California Current, North Atlantic and Northeast Pacific that used relative abundances were likely affected by issues when correlating compositional
data, and we hypothesize that stronger correlations would be observed had quantitative approaches been employed. In particular, dinoflagellates displayed relatively low correlations with relative abundances (Kramer et al., 2024b; Catlett et al., 2023), but the strength of these correlations dramatically improved by using quantitative approaches here (Fig. 2A). For cyanobacterial abundances, pigment and transcriptional variability likely contributed to weaker correlations when using absolute abundances. Smaller filter pore sizes such as those obtained by combusting GF/F filters may also need to be
considered when drawing these comparisons in future studies (Nayar and Chou, 2003)

The existence of relationships among these metrics also opens the door to several applications beyond simply assessing phytoplankton abundances and community composition. For example, PDRs can be examined in detail with not only the whole phytoplankton community but also within individual phytoplankton groups. By doing so, we show that the PDR for the whole

community in the region is negative rather than the expected unimodal distribution (Fig. 4A). By partitioning biomass into separate groups and leveraging the high resolution provided by DNA, we show that this negative trend is driven by dinoflagellates while diatoms largely show the opposite trend (Figs. 4G and 4H). These differences in PDRs align with the classical view of diatoms and dinoflagellates on opposite ends of the r- and K-selected continuum, while also showing that diatom richness increases under bloom scenarios rather than selecting for a small number of opportunists (Margalef, 1978).

The results also suggest that a shift to a more stratified less productive regime from climate change may support a more diverse phytoplankton community, but one that has low diatom diversity.

Increases in diatom and dinoflagellate pigments also align with increases in genera that contain harmful bloom-forming taxa (Fig. 5 and S12). With the harmful diatom genus *Pseudo-nitzschia*, gene expression for toxin biosynthesis was also often

elevated under higher fucoxanthin concentrations (Fig. 5E). Although increases in these genera does not always imply that there is toxin production, the detection of these pigments, and in particular, the substitution of remotely sensed chlorophyll *a* for phytoplankton pigments will likely improve HAB detection and forecasts. Increases in pigments also corresponds to higher expression of genes specific to their respective group besides with Chlorophytes. These genes fall into diverse metabolic categories, and as connections between gene expression and rate processes emerge as done with quantitative protein

measurements (McCain et al., 2024), the detection of pigments may aid the inference of group-specific metabolic activity and support the development of ESMs.

**Data availability**

Phytoplankton pigment and flow cytometry data are available on CCE-LTER Datazoo: https://oceaninformatics.ucsd.edu/datazoo/catalogs/ccelter/datasets. The DNA and RNA sequence data reported in this study

have been deposited in the National Center for Biotechnology (NCBI) sequence read archive under the BioProject accession numbers PRJNA555783, PRJNA665326, and PRJNA804265.

**Author contributions**

RHL, RG, and AEA designed the study approach. RHL, AJR, SMW, AS, and HZ performed the nucleic acid sample collection, processing, and DNA-based analyses. RHL and PV analyzed the metatranscriptomics data. MRL provided the flow cytometry

data. RHL, RG, and MRS analyzed the phytoplankton pigment data. RHL, ADB, and AEA synthesized the results. RHL performed the statistical analyses and wrote the manuscript with input from all authors.



**Competing interests**

The authors declare that they have no competing interests

**Acknowledgements**

We thank the CCE-LTER scientists, Shonna Dovel and Megan Roadman, and the captains and crews of all research vessels involved. We also thank Mark Ohman and Kathy Barbeau as former and current Lead PIs of the California Current Ecosystem Long-Term Ecological Research (CCE-LTER) site. Zoltán Füssy also assisted in the submission of this manuscript. This work was supported by the National Oceanic and Atmospheric Administration grants NA15OAR4320071 and NA19NOS4780181 (to A.E.A), the National Science Foundation grants OCE-1026607 (to R.G. and M.R.L.), OCE-1637632 (to R.G. and M.R.L.),

OCE-2224726 (to A.E.A. and M.R.S.), and DGE-1650112 (Graduate Research Fellowship to R.H.L), the Gordon and Betty Moore Foundation grant GBMF3828 (to A.E.A.), and the Simons Foundation Collaboration on Principles of Microbial Ecosystems (PriME) grant 970820 (to A.E.A.). R.H.L. also received partial support from the LaVerne Noyes Foundation.

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
