# Peer review of "Relationships between phytoplankton pigments and DNA- or RNAbased abundances support ecological applications"

_EGUsphere, 2024_

## Referee Comment (RC1)

**General comments**

Marine phytoplankton pigments determined via HPLC analysis have been extensively used to develop and validate remote sensing algorithms for determining the specific abundance of phytoplankton groups, becoming a reference metric. The study under review aims to compare phytoplankton pigments with measurements of DNA-based metabarcoding and mRNA abundances via metatranscriptomics. This study seeks to determine to what extent existing datasets of DNA metabarcoding and marine mRNA can be used to develop models of phytoplankton group distributions, supporting the next generation of hyperspectral satellites.

The manuscript appears well-structured and written. The abstract and title reflect the content accurately, and the references are appropriate. This is interesting work by comparing the significant number of samples. Even though such comparisons has been done previously, this cover different geographical locations and explored quantitative methods that express both 18S rDNA and total mRNA as concentrations. This work merits to be published after a series of minor weaknesses that are reported here will have to be addressed. These involve the following:

1) The scientific methods and assumptions are clearly outlined regarding DNA, mRNA techniques, primary productivity, and flow cytometry. However, there is a notable lack of detailed description of the HPLC pigment analysis methods, and the relevance of flow cytometry in the context of the presented work is not well explained.
2) The results are sufficient to support the interpretations and conclusions, and the description of experiments and calculations is sufficiently complete and precise. However, the discussion and conclusion sections lack a connection to the potential satellite applications of the results presented.

**Specific comments**

Introduction:

The Introduction is well-structured, and the database is adequately described and appropriate in terms of its representativeness for the study presented.

Materials and Methods:

In the Methods section, substantial attention is devoted to describing the DNA metabarcoding and mRNA methods and analyzing their results. The cytometry method is adequately described. However, the HPLC method is not described at all: the analytical procedure applied, the pigments measured, and the sample pre-treatment method are not mentioned. Additionally, there is no discussion of the uncertainty associated with the pigment measurements. It is unclear how the composition of pigments, such as chlorophyll $a$, is calculated. Furthermore, diagnostic pigments are only briefly mentioned in line 315. The abbreviation for the pigments are not clarified (i.e., Total Chlorophyll a) and uniformly used and need to be revised through all the text.

The description of the HPLC method references the Phytoclass (line 165), but its relevance to the article is no further discussed except using for the fig. 4 and fig. S4. On the other hand, at line 329 the text, there's a reference to CHEMTAX (but no link with the phytoclass). There is also no rationale provided for limiting the analysis to diagnostic pigments alone.

Another point to clarify is when cytometry data is used: an explanation of the added value of this information should be included (e.g., why cytometry is important for the *Prochlorococcus;* lines 305 and 408 but not elsewhere).

The seasonal variation is presented but not further discussed in the follow session

Results and Discussion:

Finally, both the abstract and introduction mention the potential use of this study to support the development and validation of remote sensing products. However, additional explanation on how could be realized should be added. Specifically, potential models of pigment concentrations for remote sensing of harmful algal blooms or phytoplankton community structure are not adequately explored.

Conclusion:

This conclusion is comprehensive, presenting the study's findings effectively and connecting them to broader ecological and methodological implications. However, the text could be improved for clarity, conciseness, and better flow (i.e., moving between themes like biases, PDRs, HABs, and remote sensing lack sometimes of clear transitions)Some ideas, such as the limitations of 18S rRNA gene copy number variability and the importance of quantitative approaches, are repeated multiple times, making the text unnecessarily long. jumping between themes like biases, PDRs, HABs, and remote sensing without clear transitions

**Technical corrections**

Line 115-120: capture/capturing used 3 times in few sentences: consider to use synonym

Line 160. The HPLC acronym was already introduced in line 57

Line 315 the Fuco, Perid, etc… acronyms were introduced already at line 285

Line 321 "correlation", consider to specify Pearson correlation

Line 364 "example, reduced light availability may lead to cellular increases in accessory pigments" please specify if this included accessory pigments used in the present study.

**fig 4.** Total Chl a in A and D is different from and Chl a of E-J ?

---

## Referee Comment (RC3)

**General comments:**

This manuscript addresses how to interface phytoplankton observations across many different lenses (e.g., metabarcoding, metatranscriptomics, HPLC, flow cytometry, biogeochemical rate measurements), a goal that has remained elusive due to differences in absolute quantification of organisms and relative abundances stemming from the compositional nature of molecular datasets. The authors circumvent this by using quantitative techniques, such as with the use of internal standards, to move beyond relative abundances with their molecular efforts. This allows them to complement other approaches like flow cytometry and HPLC used to measure pigment concentrations to reveal significant correlations between different eukaryotic and cyanobacteria phytoplankton groups across these different methodologies. By integrating the different approaches, they have further leveraged these relationships to interpret mechanisms setting the ecological patterns (e.g., productivity-diversity relationships, harmful algal bloom composition) in a dynamic upwelling region across both spatial and temporal dimensions.

Furthermore, since HPLC-measured pigments are routinely used to develop and validate remote sensing observation, including emerging high resolution hyperspectral remote sensing reflectance data, the authors highlight the importance of comparing phytoplankton pigments to alternate metrics, e.g., metabarcoding and metatranscriptomics, of phytoplankton community composition (PCC). The positive correlations between HPLC and molecular based PCC observed in this study are helpful in establishing the usefulness of using molecular data to further help validate global phytoplankton community structure being observed by remote sensing algorithms and developing improvements with Earth system models (ESMs).

In general, I find the authors did a nice job structuring the manuscript, building their arguments, and supporting their findings in context of what has been discussed in literature. The overall content and important take home messages are also clearly articulated. However, I think section 3.2.3 could use a bit more explicit discussion guiding how to interpret the results highlighted here and create a stronger link to how ESMs might use these results (or perhaps we should simply focus on the patterns observed as another validation reference for ESMs?).

Importantly, since so many of the relationships and ecological patterns discussed throughout the paper rely on various statistical analyses, I would strongly urge the authors to update the "Statistics" section in the methods and provide some justification for choosing Pearson correlation instead of Spearman correlations for this study (see more specific comments below for general guidelines that might be helpful). Lastly, there were several different sequencing platforms used for the various libraries prepared for metabarcoding and metatranscriptomics work – please address whether there are any biases or concerns comparing across all the different platforms (e.g., did you use unique

dual indexing pooling combinations to minimize index hopping with the NovaSeq 6000 platform).

**Specific comments:**

Figure 1: Panel D – I'm a little confused by the y-axis scale for nitrate concentrations. I think you are trying to highlight the often very low (<0.5 µM) concentrations on the same range as values as high as 20+ µM but the scaling seems a bit unorthodox. The intervals between values don't signify the same thing so is there a way to clarify that (perhaps in the figure legend)?

**Methods**

Section 2.4 & 2.5: It doesn't seem that any mock communities were used in the library prep, is that right? Please address how mock communities could also improve the quantitative assessment of this study (e.g., see conclusions from Lamb et al., 2018 - https://doi.org/10.1111/mec.14920).

For the use of Parada et al., 2016 primer set, were the 18S sequences discarded and solely the 16S sequences were denoised into ASVs? If yes, perhaps mention this – it seems to tally with your choice of removing all eukaryotic chloroplast and mitochondrial ASVs from the 16S fraction of this data (lines 206 – 207).

Lines 211 – 212: In previous method section (2.3), only the addition of *S. pombe* is mentioned so please reconcile that before introducing this step of dividing by ratio of an additional internal standard of *T. thermophilus*.

Section 2.7 Statistics: Please expand upon this section to highlight the different functions and any parameters that were modified from their default setting when using the function to carry out various analyses such as Shannon H' index, GAMs, Pearson correlations, linear regression on residuals, etc. For instance, "GAMs were fit using the function 'gam (y~s, method = "REML")' from the mgcv package v1.9-1 (Wood, 2017)." Furthermore, the interpretations and discussion rely heavily on Pearson correlations – please add some justification for why this method was chosen over others, i.e., Spearman rank-correlations. For datasets that follow a bivariate normal distribution, Pearson correlations are useful to measure linear relationships (not sure if you have tested for whether your datasets are normally distributed). However, if the datasets are nonnormally distributed or have relevant outliers, you might actually consider using an alternative like Spearman correlation to test for monotonic association. This could provide different interpretations, potentially stronger correlations, than what your current results indicate.

**Results and Discussion**
Lines 303 – 307: This section discussing the results of the cyanobacteria fraction of the data could be expanded a bit more. For instance, this potential dominance of *Prochlorococcus*

might align with the observed warming influence and advection of oligotrophic offshore waters into the study region as previously observed at the San Pedro Ocean Time-series (SPOT) where this was accompanied by a notable shift from cold-water ecotypes to warm-water ecotypes during 2014-2015 (Yeh and Fuhrman, 2022 - https://doi.org/10.1038/s41467-022-35551-4). Similarly, the 2015-2016 El Niño event also marked an increase in an open ocean ecotype of UCYN-A at SPOT (Fletcher-Hoppe et al., 2023 - https://doi.org/10.1038/s43705-023-00268-y) but it seems its presence and range of coverage was not detected with the cyanobacteria ASVs recovered from the samples collected in this study.

Figure 3: Consider specifying "All Cyanobacteria" on the figure's panel titles C and D to align with the description in the figure legend. And same thing for Figure S7.

Line 497: "…;however, contrary to predictions" Are there literary references to suggest that diversity and richness should be expected to be low in deep SCML samples – where/why did you have that hypothesis?

Figure 5: Are the samples highlighted in panel F only a subset of the samples from panel E? It is specified that the samples are ordered by the associated fucoxanthin concentrations, but it seems that only samples above a certain *dabA* expression threshold are included here – maybe clarify this selection criteria.

**Technical comments:**

Line 48: "Earth system*s* models" (make it as "system" – singular)

Line 137: Station 81.8 46.9 – are these two separate stations or just a unique nomenclature?

Line 481: "…for the mechanisms that underlying them." Awkward phrasing.

Line 535: Adjust to "…shown to produce DA and its production is…" You already introduced the acronym DA to represent domoic acid so you can maintain consistency this way.

Lines 542-544: Consider rephrasing the sentences to streamline the strucutre: "Dinoflagellates, including certain members in the genera *Alexandrium*, *Dinophysis*, and *Gonyaulax* and species *Gymnodinium catenatum* and *Lingulodinium polyedra*, may also cause HABs globally and in the region (Anderson et al., 2012, 2021; Trainer et al., 2010; Ternon et al., 2023)."

Lines 545 – 546: "although 39% of V4 and 55% of V9 18S copies…" Wouldn't referencing Figure S6B better point to these percentages – not sure the reference to Figure S13 here? Also, does blasting those sequences improve the taxonomic resolution to help better assess if there are potentially more HAB species which may currently be unassigned as HABs due to insufficient taxonomic resolution?

---

## Author Comment (AC1)

**Response to Referee 1**

**General comments**

Marine phytoplankton pigments determined via HPLC analysis have been extensively used to develop and validate remote sensing algorithms for determining the specific abundance of phytoplankton groups, becoming a reference metric. The study under review aims to compare phytoplankton pigments with measurements of DNA-based metabarcoding and mRNA abundances via metatranscriptomics. This study seeks to determine to what extent existing datasets of DNA metabarcoding and marine mRNA can be used to develop models of phytoplankton group distributions, supporting the next generation of hyperspectral satellites.

The manuscript appears well-structured and written. The abstract and title reflect the content accurately, and the references are appropriate. This is interesting work by comparing the significant number of samples. Even though such comparisons has been done previously, this cover different geographical locations and explored quantitative methods that express both 18S rDNA and total mRNA as concentrations. This work merits to be published after a series of minor weaknesses that are reported here will have to be addressed. These involve the following:

- The scientific methods and assumptions are clearly outlined regarding DNA, mRNA techniques, primary productivity, and flow cytometry. However, there is a notable lack of detailed description of the HPLC pigment analysis methods, and the relevance of flow cytometry in the context of the presented work is not well explained.

- The results are sufficient to support the interpretations and conclusions, and the description of experiments and calculations is sufficiently complete and precise. However, the discussion and conclusion sections lack a connection to the potential satellite applications of the results presented.

We thank the reviewer for their time and constructive feedback. We agree that much more detail is needed for the HPLC methods, and we have responded to the reviewer's more specific comment about this below. We also agree that the flow cytometry data requires further explanation and have made several revisions to do so. Specifically, we now:

- State in the Introduction on lines 122-123 "Within cyanobacteria, flow cytometry was also used to measure *Prochloroccocus* and *Synechococcus* cell abundances."
- Describe in the beginning of the Flow Cytometry section in Materials and Methods on lines 292-293: "To measure *Prochloroccus* and *Synechococcus* cell abundances with flow cytometry, providing an additional metric of comparison…"

We also agree that the discussion and the conclusions require more connection to the satellite applications. In response we now state:

- Results and Discussion, lines 487-491: "As phytoplankton pigments directly impact remote sensing reflectance spectra, these results further support that potential models for phytoplankton pigment concentrations via remote sensing may be able to provide comparable global estimates of PCC (Kramer et al., 2022). While HPLC pigments are used validating remote-sensing algorithms, these results also suggest that the absolute

abundances of DNA or RNA may be useful metrics to validate for model development of PCC as well."

- Results and Discussion, Section 3.2.2 Monitoring and forecasting harmful algal blooms, lines 601-605: "In the California Current region, *Pseudo-nitzschia* abundances and DA production are forecasted by the California Harmful Algae Risk Mapping (C-HARM) system, which uses satellite remote-sensing data and a regional ocean circulation model. Specifically, the remote-sensing data used as inputs are chlorophyll *a* concentrations and reflectance at 488 and 555 nm from the S-NPP NOAA VIIRS instrument. As Fuco offers greater specificity for diatoms, substituting Fuco for chlorophyll a may improve model predictions, particularly if *Pseudo-nitzschia* is a dominant diatom overall."

- Conclusion, lines 685-691: "By integrating phytoplankton pigments with quantitative abundances of 18S rRNA genes and total mRNA via metabarcoding and metatranscriptomics respectively, we demonstrate that diagnostic pigments for specific eukaryotic phytoplankton groups correlate with both their DNA- and RNA-based abundances. Although there are inherent biases associated with each of these measurements, their relationships suggest that they are comparable and may all individually be useful for validating potential models of PCC from hyperspectral remote sensing reflectance with satellites such as PACE. These relationships also suggest that the potential development of models for remotely sensed pigment concentrations will provide reasonable estimates for the abundances of different phytoplankton groups (Kramer et al., 2022)."

**Specific comments**

Introduction:

The Introduction is well-structured, and the database is adequately described and appropriate in terms of its representativeness for the study presented.

Thank you.

Materials and Methods:

In the Methods section, substantial attention is devoted to describing the DNA metabarcoding and mRNA methods and analyzing their results. The cytometry method is adequately described. However, the HPLC method is not described at all: the analytical procedure applied, the pigments measured, and the sample pre-treatment method are not mentioned. Additionally, there is no discussion of the uncertainty associated with the pigment measurements. It is unclear how the composition of pigments, such as chlorophyll a, is calculated. Furthermore, diagnostic pigments are only briefly mentioned in line 315. The abbreviation for the pigments are not clarified (i.e., Total Chlorophyll a) and uniformly used and need to be revised through all the text.

We have now expanded the methods text to describe the additional information requested by reviewer, including the sample extraction, analytical procedure, pigments measured that were used in this study, and estimates of uncertainty. We also provide citations that describe the Horn Point Laboratory (HPL) method, which was the method used here, in detail (Hooker, 2005; Van Heukelem and Thomas, 2001).

> With respect to describing the diagnostic pigments, they are first mentioned in the introduction on lines 60-62: "certain accessory pigments can be used as proxies, or diagnostic pigments, to determine the abundances of specific groups, even though some of these pigments are shared among groups (Jeffrey et al., 2011; Kramer and Siegel, 2019).

> We also now clarify the use of diagnostic pigments in the Materials and Methods on lines 191-192: As described later in the results and discussion, several of these pigments are diagnostic pigments for certain phytoplankton lineages."

> We also list the diagnostic pigments, which taxa they are diagnostic for, and their abbreviations on lines 357-361.

> As suggested, we have also modified the text to now use the pigment abbreviations more uniformly throughout the text, except in a small number of specific circumstances where we feel that it would be clear to use the full pigment name.

The description of the HPLC method references the Phytoclass (line 165), but its relevance to the article is no further discussed except using for the fig. 4 and fig. S4. On the other hand, at line 329 the text, there's a reference to CHEMTAX (but no link with the phytoclass). There is also no rationale provided for limiting the analysis to diagnostic pigments alone.

> We have now modified this methods text to describe the relevance of also analyzing the pigments with phytoclass on lines 193-195: "In addition to use diagnostic pigments directly, taxon-specific contributions to TChla concentrations based on the aforementioned pigments were determined with phytoclass v2.0.0."

> With respect to CHEMTAX, it is only mentioned to describe the results of another study where it was used (lines 371-373). We mention that is another chemotaxonomic approach; therefore, we do not believe that it is necessary to further describe.

> As suggested, we also now provide rationale for solely examining diagnostic pigments on lines 192-193: "Although other pigments are measured with HPLC, they do not provide as much specificity as the diagnostic pigments used here; therefore, they were not included in the analysis (Kramer and Siegel, 2019)."

Another point to clarify is when cytometry data is used: an explanation of the added value of this information should be included (e.g., why cytometry is important for the *Prochlorococcus;* lines 305 and 408 but not elsewhere).

> We now clarify this methods text by introducing the section with the following (lines 292-293): "To measure *Prochloroccus* and *Synechococcus* cell abundances with flow cytometry as an additional metric of comparison…"

The seasonal variation is presented but not further discussed in the follow session

> Our intention in mentioning that these samples were collected among different seasons is to describe that potential seasonal variability is accounted for within our primary objective of comparing HPLC pigments to DNA- and RNA-based abundances. Our goal in this manuscript is not to examine potential seasonal patterns in phytoplankton pigments or groups in the region, and we

believe that such an analysis is beyond the scope of this manuscript. As a result, we have not made additional revisions in response to this comment.

Results and Discussion:

Finally, both the abstract and introduction mention the potential use of this study to support the development and validation of remote sensing products. However, additional explanation on how could be realized should be added. Specifically, potential models of pigment concentrations for remote sensing of harmful algal blooms or phytoplankton community structure are not adequately explored.

Please see our response above to the reviewer's general comment about strengthening the connections to satellite applications. In particular, we describe revisions made that further describe how models of pigment concentrations can aid understanding phytoplankton community structure and harmful algal blooms.

Conclusion:

This conclusion is comprehensive, presenting the study's findings effectively and connecting them to broader ecological and methodological implications. However, the text could be improved for clarity, conciseness, and better flow (i.e., moving between themes like biases, PDRs, HABs, and remote sensing lack sometimes of clear transitions)Some ideas, such as the limitations of 18S rRNA gene copy number variability and the importance of quantitative approaches, are repeated multiple times, making the text unnecessarily long. jumping between themes like biases, PDRs, HABs, and remote sensing without clear transitions

We agree with the reviewer that the conclusion could be restructured and revised for clarity, and we done so to make these improvements. Please see the conclusions text in the revised manuscript.

**Technical corrections**

Line 115-120: capture/capturing used 3 times in few sentences: consider to use synonym

We have rephrased this text to avoid repeated use of derivates of capture (lines 120-134).

Line 160. The HPLC acronym was already introduced in line 57

We have removed the unabbreviated text (line 180).

Line 315 the Fuco, Perid, etc… acronyms were introduced already at line 285

Although the abbreviations for fucoxanthin and 19'-hexanoyloxyfucoxanthin were previously introduced, we feel that reintroducing them here aids the reader since the context of the sentence is to list all the pigments used in this study, their abbreviations, and their assignment as diagnostic for certain taxonomic groups.

Line 321 "correlation", consider to specify Pearson correlation

We now specify that the correlations are Pearson correlations here (line 364).

Line 364 "example, reduced light availability may lead to cellular increases in accessory pigments" please specify if this included accessory pigments used in the present study.

We now state that "reduced light availability may lead to cellular increases in **all** accessory pigments **examined here**" on lines 407-408.

fig 4. Total Chl a in A and D is different from and Chl a of E-J ?

Yes, panels A-D show total chlorophyll *a* concentrations whereas panels E-J show taxon-specific chlorophyll *a* concentrations estimated from phytoclass. To make this clearer, we have modified the x-axis of panels E-J to now state "Taxon-specific Chl *a*," and the caption now species that the panels are showing "diversity of individual phytoplankton groups against their taxon-specific chlorophyll *a* concentrations estimated with phytoclass."

**References**

Hooker, S. B.: The second SeaWiFS HPLC analysis round-robin experiment (SeaHARRE-2), National Aeronautics and Space Administration, Goddard Space Flight Center2005.

Kramer, S. J. and Siegel, D. A.: How Can Phytoplankton Pigments Be Best Used to Characterize Surface Ocean Phytoplankton Groups for Ocean Color Remote Sensing Algorithms?, J Geophys Res Oceans, 124, 7557–7574, https://doi.org/10.1029/2019JC015604, 2019.

Kramer, S. J., Siegel, D. A., Maritorena, S., and Catlett, D.: Modeling surface ocean phytoplankton pigments from hyperspectral remote sensing reflectance on global scales, Remote Sens Environ, 270, 112879, https://doi.org/10.1016/j.rse.2021.112879, 2022.

Van Heukelem, L. and Thomas, C. S.: Computer-assisted high-performance liquid chromatography method development with applications to the isolation and analysis of phytoplankton pigments, Journal of Chromatography A, 910, 31–49, 2001.

---

## Author Comment (AC2)

**Response to Referee 2**

General comments:

This is a valuable and timely contribution for the PACE era. The use of quantitative omics is very much a step in the right direction, in my opinion, and the strong and improved correlations observed between quantitative omics and pigment data are reassuring.

Overall, I found the writing and presentation to be of a high standard and the collection of data over multiple years and seasons is a commendable feat.

The authors have done a good job considering environmental gradients and seasonal change, but it feels like an opportunity was missed by not doing a more granular analysis of surface vs. deep communities if the data are in hand.

I found the ecological application 'case studies' to be a mixed bag. On the one hand, Section 3.2.1 that considers ecological assessments shows a lot of promise of combining omics and pigment data to improve understanding of ecological processes. On the other hand, I'm not persuaded by the suggestion of Section 3.2.2 that quantitative omics, as presented in this manuscript, can improve the monitoring and forecasting of harmful algal blooms. But, I hope the authors can change my mind about this. Finally, Section 3.2.3 suggests that quantitative omics may lead to better biogeochemical and metabolic rate estimates. I think this section could be improved by considering further the substantial caveats and current limitations to this potential application.

Finally, the Methods section is missing important details in places, which are described in the Specific comments below.

> We thank the reviewer for their time and constructive feedback. As the reviewer describes these general comments in greater detail below, we have responded to them in addition to their other comments in line.

Specific comments:

Fig. 1 Figure caption; L129-130: Consider adding "regions" after (18S-V9 (blue)"

> We have modified this text to now read: "Relative abundances of different phytoplankton groups using 18S-V4 **rRNA gene** (red), 18S-V9 **rRNA gene** (blue), or transcript (metaT, green) abundances"

Methods:

L137-140: I understand what the authors are trying to say, but this is an awkwardly worded sentence. Perhaps along the lines of "These data represent only a subset of the on-going NOAA-CalCOFI Ocean Genomics (NCOG) time series and are restricted to samples where quantitative approaches for DNA and RNA were employed concurrently with phytoplankton pigments samples (no DNA samples from 2017 and only RNA samples from 2017-2020; James et al., 2022)."

> We have revised this text as suggested (lines 148-151).

LN150-158: The primary productivity section is sparse and could benefit from additional details.

We agree with the reviewer and have made several revisions detailed below to clarify how the primary productivity measurements were performed.

How were the sampling depths determined? Was there a bio-optics CTD cast to determine the light extinction coefficient prior to sampling for [14]-C incubations, and the sampling depths were then chosen to match the degree of attenuation of the neutral-density screens used for the deckboard incubation?

We now describe the sampling depths and light levels on lines 164-167: "seawater was collected from six depths representing 56%, 30%, 10%, 3%, 1%, and 0.3% surface light levels shortly before local apparent noon. Light levels were estimated with a Secchi disk with the assumptions that the 1% light level is three times the Secchi depth and that the extinction coefficient is constant."

This approach is used to maintain consistency within the time series, as this protocol was implemented in 1984, prior to the CalCOFI program's transition to using a CTD rosette.

What was the specific activity, concentration of radioactivity added to the sample bottles, and supplier of the $NaH^{14}CO_3$ used?

The specific activity varies by cruise depending on the activity provided by the supplier, MP Biomedicals LLC. We have updated the methods text to be more detailed and include this information on lines 168-170: "Bottles were then inoculated with a 200 µL solution containing $NaH^{14}CO_3$ that was prepared by diluting 50 mL of $NaH^{14}CO_3$ (approximately 50-57 µCi mmol$^{-1}$; MP Biomedicals, LLC) with 350 mL of 2.8 mM $Na_2CO_3$ and then adjusting the pH to ~10 with 1 N NaOH (Fitzwater et al., 1982)."

I believe 'HA' is a Millipore-specific designation for filter type. I suggest mentioning the material of the filter (i.e., mixed cellulose esters (MCE) membrane) for those reader who are not familiar with them.

We have modified this text as suggested to now read (lines 171-172): "Following incubation, samples were filtered onto 0.45 µm mixed-cellulose ester filters (type HA, Millipore)"

Was the incubation time a constant between seasons? Multiplying by 1.8 to obtain 24 h productivity implies that the incubation time was ~ 13.3 h, but you state also that incubations were performed between local noon and civil twilight, which presumably varies seasonally.

The reviewer is correct that the incubation time varies seasonally with the length of daylight. As the incubations occur from local noon to civil twilight, the resulting measurement is half light-day productivity. Previous direct comparisons between half light-day productivity incubations and 24-hour incubations on CalCOFI and other regional cruises were made by Eppley (1992). As shown below in Figures 8 and 9 from this publication, the 24 hour values are approximately 1.8 times the half light-day values.

[Figure]

FIG.8. Comparison of half-day photosynthetic production, measured from noon to sunset, with that measured 24 hours on SCBS cruises 24 and 25. The regression line is 24h production = 1.80 (half-day production) + 0.62. The regression explained 91% of the variability, i.e. $r^2 = 0.907$.

[Figure]

FIG.9. As in Fig.8, except the experiments were done on several CalCOFI cruises and values are per area rather than per volume. The slope of the line is 1.8 from Fig.8.

We have updated this text to clarify this. It now reads on lines 173-176: "As the incubations occurred from local noon until civil twilight, half light-day productivity at each depth was calculated by averaging the two light bottles corrected with the dark uptake bottle. Half light-day productivity was then multiplied by 1.8 to obtain 24 hour productivity as determined by Eppley (1992)."

How closely did the collection depths of productivity and DNA samples match? "Closest" is vague.

We have updated this text to clarify the differences in depths between productivity and DNA samples (lines 176-178): "When comparing productivity to diversity from DNA, samples from the entire NCOG dataset (2014 to 2020) that were within 20 m of productivity samples were used ($n =$ 434). The average vertical distance between DNA and productivity samples was 1.79 m."

L160-165: Analytical details of the HPLC pigment analyses are entirely missing – you skip directly from sample collection to Phytoclass taxonomic analyses. I suggest adding a citation to the analytical method at the very least.

We have revised this text to now include details of the HPLC analysis with citations that further detail the method (lines 182-191):

"Once completed filtering, the filters were carefully folded in half, blotted on a paper towel to remove excess water, and stored in 2 mL cryovials in liquid nitrogen until analysis at the Horn Point Analytical Services Laboratory at the University of Maryland with the HPL method as described in Hooker (2005). Briefly, filters were extracted in 95% acetone and sonicated on ice for 30 s with an output of 40 W. Samples were then clarified by filtering them through a HPLC syringe cartridge filter (0.45 µm) and a glass-fiber prefilter. Extracts were then analyzed with an automated HP 1100 HPLC system with external calibration standards that were either purchased or isolated from naturally occurring sources as described in Van Heukelem and Thomas (2001). The pigments that were measured and used here are Peridinin, 19'-butanoyloxyfucoxanthin, fucoxanthin, neoxanthin, prasinoxanthin, violaxanthin, 19'-hexanoyloxyfucoxanthin, alloxanthin, zeaxanthin, divinyl chlorophyll a, and TChla. The precision including filter extraction and analysis of TChla is estimated to be 5.1%."

L255: Flow cytometery. It appears that the only flow cytometry data that is presented is for cell abundances for *Prochlorococcus* and *Synechococcus* in Fig. S2. As such, you may wish to consider moving this portion of the methods to the supplementary information. I suggest also that you tailor the description of the methods to explain how *Prochlorococcus* and *Synechococcus* were differentiated and gated.

Cell abundances from flow cytometry for *Prochlorococcus* are included in Figure 3E and 3F. Since these data are in a main figure, we would prefer to keep the associated methods text in the main Materials and Methods rather than the Supplementary Information. With respect to differentiating *Prochlorococcus* and *Synechococcus*, we also now add a citation for Monger and Landry (1993) which details the flow cytometry methods used here.

L264-266: I'm confused about the use of side- versus forward-scatter. Were forward *and* size-scatter signatures used to estimate the size of *Prochlorococcus* and *Synechococcus*?

Yes, we now state on lines 301-304: "The optical filter configuration distinguished ***Prochlorococcus*** **and *Synechococcus* populations** based on chlorophyll a (red fluorescence, 680 nm), phycoerythrin (orange fluorescence, 575 nm), DNA (blue fluorescence, 450 nm), and forward and 90° side-scatter signatures."

Results and Discussion

LN286: As you're discussing the range here, I suggest reporting the range for Fuco, rather than the maximum concentration, as you don't specify in the previous sentence what pigments were measured at the lowest concentrations (and the log scale used in Fig. 1E makes this difficult to discern by eye).

> We have rephrased this text to not only be more specific in reporting the range Fuco and other pigments used here. We now state on lines 327-329:

> > "Fucoxanthin (Fuco) exhibited the greatest range with concentrations ranging from 0.001 to 6.81 µg $L^{-1}$. While 19'-hexanoyloxyfucoxanthin (HexFuco) concentrations ranged from 0.005 to 1.170 µg $L^{-1}$, all other pigment concentrations examined here were always less than 0.697 µg $L^{-1}$."

L364-372: The authors state that samples were collected from both the near-surface and SCML. It's reasonable to assume that the two communities may differ systematically in terms of both community composition and photophysiology, as the authors discuss. Were samples from these two depths pooled for all analyses? If so, how did the sample numbers differ between the two depths, potentially skewing the aggregated results?

> Yes, the samples from all depths were pooled for analysis, reflecting a wide range of conditions and light histories. To clarify that there were similar sample numbers among depth categories, we now state in the Methods on lines 158-159: "For the DNA samples, 219 were from the near-surface (0 – 14 m, mean = 10 m), and 198 were from the SCML (18 – 130 m, mean = 53 m)."

> This added text also clarifies that the sampling depths for the SCMLs are wide ranging, since SCMLs in the region generally become deeper from the nearshore environment to the offshore environment along with the nitracline. Considering the amount of variability captured by pooling all samples, we believe that the strength of the correlations is a strength of our analysis; however, we agree with the reviewer that more attention should be paid to surface versus deeper communities as detailed in response to the next comment.

The authors have done a good job considering environmental gradients and seasonal change, but it seems like a missed opportunity to not extend these analyses to surface vs. deep communities if the data are in hand. Have the authors performed the regressions shown in Fig. 2 on low- vs high-light binned samples? Even if they are do not differ from each other, this is useful information and could be included in the supplementary information.

> As suggested, we have examined the relationships between pigments and DNA-based abundances for surface and SCML communities separately (Figs. S6 and S9). We also describe these results on lines 417-425:

> > "To further examine the effects of light and depth on these relationships, separate correlations were performed with the absolute abundances from near-surface (≤ 14 m) or SCML (≥ 18 m) samples. For chlorophytes, cryptophytes, diatoms, dinoflagellates, and prymnesiophytes, the strength of the correlations were similar or higher for both depth categories when separated compared to all samples combined (Figs. 2 and S6). Only pelagophyte correlations were consistently lower when separated, albeit the differences were relatively minor ($r = 0.65 – 0.70$ versus $r = 0.74 - 0.75$). Linear regressions also displayed similar results between depth categories,

except for chlorophytes and prymnesiophytes, where the slope of the regressions were 46-49% lower with SCML samples. In these cases, pigment concentrations exhibited a reduced range, where concentrations were elevated wat lower DNA abundances but reached similar concentrations at higher DNA abundances, indicating that pigment concentrations for these taxa are elevated under lower abundance regimes within the SCML."

And on lines 465-467:

"Although separating near-surface and SCML samples showed stronger correlations than all samples together except for all cyanobacteria and Zea ($r$ = 0.55), the relationships were still weaker than with relative abundances (Fig. S9)."

Section 3.2.1 This was a useful illustrative example of how to combine the strengths of 'classical' HPLC data and DNA-based approaches to better understand phytoplankton ecology. I enjoyed reading it.

Thank you.

L500-509: The use of recycled nutrients (a low f-ratio in the case of nitrogen, a low Fe-ratio in the case or iron) should be included as a general strategy.

We agree and have now added a sentence to mention recycled nutrients and nitrogen fixation as means to sustain phytoplankton under oligotrophic conditions. This section now reads (lines 565-569):

"The low productivity and diversity end of unimodal distributions have also been attributed to selective grazing with the dominance of a few slow-growing nutrient specialists (Vallina et al., 2014). As diversity and richness instead remained high, many phytoplankton taxa, particularly dinoflagellates, appear to coexist within low productivity regimes. With low nutrient availability as inferred by deeper nitracline depths (Fig. 4D), diverse phytoplankton taxa within these regimes may be sustained by recycled nutrients including nitrogen and iron as well as nitrogen fixation (Boyd et al., 2017; Zehr and Ward, 2002)."

Section 3.2.2 This section was not nearly as compelling as the previous one. In effect, you argue that if fucoxanthin is detected then there are likely to be *some Pseudo-nitzschia* present, *some* of which *may* produce DA. It's also quite possible for fucoxanthin concentrations to vary independently of *Pseudo-nitzschia* abundance. It's not clear how this improves the current state of knowledge or improves the monitoring and forecasting of harmful algal blooms, even within an intensively sampled region like the CalCOFI survey site.

To clarify how the results of our analysis suggest that Fuco detection may aid HAB monitoring, we now describe that current HAB monitoring for *Pseudo-nitzschia* blooms in the California Current relies on remotely-sensed chlorophyll *a* and reflectance from older sensors. Since Fuco offers greater specificity for diatoms, and if *Pseudo-nitzschia* is regularly a dominant diatom, substituting chlorophyll *a* with Fuco in the model may improve accuracy. This is now stated on lines 601-605:

"In the California Current region, *Pseudo-nitzschia* abundances and DA production are forecasted by the California Harmful Algae Risk Mapping (C-HARM) system, which uses satellite remote-sensing data and a regional ocean circulation model. Specifically, the remote-sensing data used as inputs are chlorophyll *a* concentrations and reflectance at 488 and 555 nm from the S-NPP

NOAA VIIRS instrument. As Fuco offers greater specificity for diatoms, substituting Fuco for chlorophyll *a* may improve model predictions, particularly if *Pseudo-nitzschia* is a dominant diatom overall."

As we further describe in this section, *Pseudo-nitzschia* is indeed one of the most dominant diatoms, and Fuco concentrations were better predictors of *Pseudo-nitzschia* than TChla (Fig. 4). Moreover, increased Fuco concentrations generally aligned with increased expression of a critical domoic acid biosynthesis gene, *dabA*. As a result, we believe that we have shown evidence that Fuco detection may support improvements to HAB forecasts in this region, and potentially others where *Pseudo-nitzschia* is a dominant diatom; however, we recognize that there are important caveats that must be considered, such as the prerequisite that *Pseudo-nitzschia* regularly is a dominant diatom taxa. We also recognize that potential improvements to C-HARM forecasts in the California Current for *Pseudo-nitzschia* or forecasts in other regions would require additional validation. We now describe these caveats on:

- Lines 621-624: "Pending the development and implementation of models for remotely-sensed Fuco concentrations (Kramer et al., 2022), such potential improvements for *Pseudo-nitzschia* forecasts will require validation with *in situ* measurements. Moreover, the utility of remotely-sensed Fuco concentrations for *Pseudo-nitzschia* HAB monitoring would only apply to other regions where *Pseudo-nitzschia* is a dominant diatom."
- Lines 729-730: "for both HAB forecasts and the inference of phytoplankton activity, significant additional validation will be required."

Section 3.2.3 I appreciated the ambition and hope of this section. One thing to note about L587-591: the modules or subsystems of genes that predict a reaction rate are likely to be different between species. As such, it would seem to me that to make use of this correlation, not only would you need to know about abundance and expressed metabolism, but you would need to have a comparable degree of knowledge of your target species as we have for baker's yeast – an extensively studied model organism – to know which gene clusters predict rates. Would this not require intensive lab rate measurements to validate this correlation? Also, the baker's yeast correlation was achieved under steady-state conditions. Trying to accomplish this in a field study? Yikes.

We certainly agree with the reviewer that this section is highly ambitious and has important caveats that must be described. We also agree that extensive additional measurements would be required to connect pigments or transcript abundances to any rates. Our intention in highlighting the study by McCain et al. (2025), which draws relationships between proteins and rates in baker's yeast, is to introduce the concept that proteins or groups of proteins may relate to rates, rather than stating that we have demonstrated any such relationship with pigments or transcripts. We now clarify these caveats on lines 647-649, lines 679-683, and 729-730:

"The absolute quantities of certain proteins have shown promise for inferring rates of nitrite oxidation and carbon fixation (Saito et al., 2020; Roberts et al., 2024), although it is unclear if absolute transcript abundances will be able to serve a similar purpose."

"Although these relatively strong correlations between pigments and transcripts indicates that this application has potential use to infer activity, direct relationships with rates remain to be

demonstrated, and establishing these relationships would require extensive additional validation with field-based studies that integrate these measurements."

"for both HAB forecasts and the inference of phytoplankton activity, significant additional validation will be required."

We believe that with this added text, we have been cautious to not overstate our claims and are upfront about the speculative nature of this discussion. Despite this high uncertainty, we still believe that it is useful to present these relationships and describe these concepts with the data in hand, even if it is ultimately shown in future studies that these connections cannot be made.

Technical corrections:

Fig. 2, column B: Y-axis number format is sometimes in scientific format, sometimes not. I suggest keeping this consistent amongst panels.

As suggested, we have modified this figure to have consistent y-axis labeling.

L481: "The observations of unimodal PDRs have led to hypotheses for the mechanisms that underlying them." Remove 'that' before 'underlying'.

"That" has been removed.

**References**

Boyd, P. W., Ellwood, M. J., Tagliabue, A., and Twining, B. S.: Biotic and abiotic retention, recycling and remineralization of metals in the ocean, Nat Geosci, 10, 167, 10.1038/ngeo2876, 2017.

Eppley, R. W.: Chlorophyll, photosynthesis and new production in the Southern California Bight, Prog Oceanogr, 30, 117–150, https://doi.org/10.1016/0079-6611(92)90010-W, 1992.

Fitzwater, S. E., Knauer, G. A., and Martin, J. H.: Metal contamination and its effect on primary production measurements1, Limnology and Oceanography, 27, 544–551, 10.4319/lo.1982.27.3.0544, 1982.

Hooker, S. B.: The second SeaWiFS HPLC analysis round-robin experiment (SeaHARRE-2), National Aeronautics and Space Administration, Goddard Space Flight Center2005.

Kramer, S. J., Siegel, D. A., Maritorena, S., and Catlett, D.: Modeling surface ocean phytoplankton pigments from hyperspectral remote sensing reflectance on global scales, Remote Sens Environ, 270, 112879, https://doi.org/10.1016/j.rse.2021.112879, 2022.

McCain, J. S. P., Britten, G. L., Hackett, S. R., Follows, M. J., and Li, G.-W.: Microbial reaction rate estimation using proteins and proteomes, ISME J, wraf018, 10.1093/ismejo/wraf018, 2025.

Monger, B. C. and Landry, M. R.: Flow Cytometric Analysis of Marine Bacteria with Hoechst 33342, Appl Enviro Microbiol, 59, 905–911, doi:10.1128/aem.59.3.905-911.1993, 1993.

Roberts, M. E., Bhatia, M. P., Rowland, E., White, P. L., Waterman, S., Cavaco, M. A., Williams, P., Young, J. N., Spence, J. S., Tremblay, J.-É., Montero-Serrano, J.-C., and Bertrand, E. M.: Rubisco in high Arctic tidewater glacier-marine systems: A new window into phytoplankton dynamics, Limnol Oceanogr, 69, 802–817, https://doi.org/10.1002/lno.12525, 2024.

Saito, M. A., McIlvin, M. R., Moran, D. M., Santoro, A. E., Dupont, C. L., Rafter, P. A., Saunders, J. K., Kaul, D., Lamborg, C. H., and Westley, M.: Abundant nitrite-oxidizing metalloenzymes in the mesopelagic zone of the tropical Pacific Ocean, Nat Geosci, 13, 355–362, 2020.

Vallina, S. M., Follows, M. J., Dutkiewicz, S., Montoya, J. M., Cermeno, P., and Loreau, M.: Global relationship between phytoplankton diversity and productivity in the ocean, Nat Commun, 5, 4299, 10.1038/ncomms5299, 2014.

Van Heukelem, L. and Thomas, C. S.: Computer-assisted high-performance liquid chromatography method development with applications to the isolation and analysis of phytoplankton pigments, Journal of Chromatography A, 910, 31–49, 2001.

Zehr, J. P. and Ward, B. B.: Nitrogen Cycling in the Ocean: New Perspectives on Processes and Paradigms, Applied and Environmental Microbiology, 68, 1015–1024, doi:10.1128/AEM.68.3.1015-1024.2002, 2002.

---

## Author Comment (AC3)

**Response to Referee 3**

**General comments:**

This manuscript addresses how to interface phytoplankton observations across many different lenses (e.g., metabarcoding, metatranscriptomics, HPLC, flow cytometry, biogeochemical rate measurements), a goal that has remained elusive due to differences in absolute quantification of organisms and relative abundances stemming from the compositional nature of molecular datasets. The authors circumvent this by using quantitative techniques, such as with the use of internal standards, to move beyond relative abundances with their molecular efforts. This allows them to complement other approaches like flow cytometry and HPLC used to measure pigment concentrations to reveal significant correlations between different eukaryotic and cyanobacteria phytoplankton groups across these different methodologies. By integrating the different approaches, they have further leveraged these relationships to interpret mechanisms setting the ecological patterns (e.g., productivity-diversity relationships, harmful algal bloom composition) in a dynamic upwelling region across both spatial and temporal dimensions.

Furthermore, since HPLC-measured pigments are routinely used to develop and validate remote sensing observation, including emerging high resolution hyperspectral remote sensing reflectance data, the authors highlight the importance of comparing phytoplankton pigments to alternate metrics, e.g., metabarcoding and metatranscriptomics, of phytoplankton community composition (PCC). The positive correlations between HPLC and molecular based PCC observed in this study are helpful in establishing the usefulness of using molecular data to further help validate global phytoplankton community structure being observed by remote sensing algorithms and developing improvements with Earth system models (ESMs).

> We thank the reviewer for their time and constructive feedback. Please see our responses inline below.

In general, I find the authors did a nice job structuring the manuscript, building their arguments, and supporting their findings in context of what has been discussed in literature. The overall content and important take home messages are also clearly articulated. However, I think section 3.2.3 could use a bit more explicit discussion guiding how to interpret the results highlighted here and create a stronger link to how ESMs might use these results (or perhaps we should simply focus on the patterns observed as another validation reference for ESMs?).

> Section 3.2.3 shows that certain pigments also strongly correlate with the expression of specific genes within the groups they are respectively diagnostic of. As described on lines 645-647, ESM parameters include biological rates and biogeochemical fluxes, and there is interest in being able to infer these rates from omics data. By showing these correlations, we posit that if connections between these transcripts and rates can be established, the relationship between transcripts and pigments also suggest that pigments can be used as a proxy. We now clarify this on lines 675-677:
>
> > "If the expression of these pathways are also found to correspond to changes in group-specific reaction rates, then the detection of these pigments with remote sensing may be useful for inferring group-specific activities, which could better constrain ESM parameters leading to more accurate predictions."

Importantly, since so many of the relationships and ecological patterns discussed throughout the paper rely on various statistical analyses, I would strongly urge the authors to update the "Statistics" section in the methods and provide some justification for choosing Pearson correlation instead of Spearman correlations for this study (see more specific comments below for general guidelines that might be helpful). Lastly, there were several different sequencing platforms used for the various libraries prepared for metabarcoding and metatranscriptomics work – please address whether there are any biases or concerns comparing across all the different platforms (e.g., did you use unique dual indexing pooling combinations to minimize index hopping with the NovaSeq 6000 platform).

With respect to the reviewer's comments about our methods for statistics, we have made revisions and responded to their more detailed comment about the topic below.

While we employed multiple sequencing platforms, only one library for 18S here was sequenced on a different platform. All of the 16S and ITS2 samples here were sequenced on the same platform within each dataset. We are confident that the error rates with Illumina sequencing are sufficiently low, and we have also used DADA2 which has been shown to be effective at correcting remaining sequencing errors (Callahan et al., 2016).

Some metatranscriptomics samples were sequencing on an Illumina HiSeq 4000 while others were sequenced on an Illumina NovaSeq 6000. The use of two platforms was unavoidable as sequencing technology has evolved over the course of the time series. However, we have no reason to believe that the change in platform has introduced any bias.

With respect to the reviewer's question about unique dual indexes (UDIs), we did use for our metatranscriptomics samples. This is now described in the methods on lines 254 and 265. Our metabarcoding libraries did not use UDIs; however, the absence of taxa not included in our mock community samples (described in the reviewers comment about mock communities below) as well as extremely few reads in sequenced PCR blanks and unused barcodes indicates that index hopping was minimal. We also believe that the multiple rounds of PCR clean up as described in our methods contribute to a reduction in index hopping as free adapters are effectively removed. We now describe this on lines 231-232:

"In addition to the mock communities, PCR blank samples and unused barcodes were also analyzed to confirm minimal index hopping."

**Specific comments:**

Figure 1: Panel D – I'm a little confused by the y-axis scale for nitrate concentrations. I think you are trying to highlight the often very low (<0.5 µM) concentrations on the same range as values as high as 20+ µM but the scaling seems a bit unorthodox. The intervals between values don't signify the same thing so is there a way to clarify that (perhaps in the figure legend)?

The nitrate concentrations are shown on a cube root scale. This is now stated in the figure legend. The reviewer is correct that we are using a cube root scale to better show the skewed distribution while still preserving zero values (below detection limit).

**Methods**

Section 2.4 & 2.5: It doesn't seem that any mock communities were used in the library prep, is that right? Please address how mock communities could also improve the quantitative assessment of this study (e.g., see conclusions from Lamb et al., 2018 -  https://doi.org/10.1111/mec.14920).

> Mock communities provided by the Fuhrman lab at USC were included in each library. We now state on lines 229-231:
>
> > "For all 16S and 18S libraries, mock communities were included as described in Yeh et al. (2021). The results from the mock community samples are shown in James et al. (2022) which validate the absence of taxon disappearance observed in previous studies."
>
> We thank the reviewer for pointing out the study by Lamb et al 2018. In that study, a meta-analysis was performed to examine the variance in slope between expected and measured relative abundances in mock communities. However, this approach does not consider some important issues. For example, the environmental communities used here are much more complex and likely contain sequences with PCR primer mismatches. Furthermore, linear regressions with relative abundances may not be an appropriate statistical approach since the data are compositional (Gloor et al., 2017).
>
> Rather than trying to assess variability in amplicon sequencing among relative abundances with mock communities and then applying that uncertainty to our environmental data, the goal of our manuscript is to compare between independent measures, i.e. HPLC pigments and 18S rRNA genes, where absolute abundances are used, avoiding issues of compositionality. We certainly appreciate the work being done to evaluate amplicon sequencing with mock communities, but as such, we believe that such an analysis is beyond the scope of this manuscript.

For the use of Parada et al., 2016 primer set, were the 18S sequences discarded and solely the 16S sequences were denoised into ASVs? If yes, perhaps mention this – it seems to tally with your choice of removing all eukaryotic chloroplast and mitochondrial ASVs from the 16S fraction of this data (lines 206 – 207).

> With the 515F-Y/926R primer set, 18S sequences are largely automatically discarded during processing by DADA2 as the sequences are too long to overlap with 300 bp paired-end sequencing. To preserve the data, the sequences would need to be artificially merged and processed separately as described in Yeh et al. (2021) and McNichol et al. (2025). However, we still removed any sequences classified as eukaryotic. This is now stated on lines 241-242: "When examining 16S relative abundances, all eukaryotic, plastid, and mitochondrial ASVs were removed."

Lines 211 – 212: In previous method section (2.3), only the addition of *S. pombe* is mentioned so please reconcile that before introducing this step of dividing by ratio of an additional internal standard of *T. thermophilus*.

> We thank the reviewer for pointing out this omission. We now state on lines 203-205: "At the start of DNA extraction during the addition of lysis buffer, 1.74 to 3.78 ng of *Schizosaccharomyces pombe* genomic DNA and 3.36 to 7.09 ng of *Thermus thermophilus* genomic DNA was added to each sample as an internal standard (Lin et al., 2019)."

Section 2.7 Statistics: Please expand upon this section to highlight the different functions and any parameters that were modified from their default setting when using the function to carry out various analyses such as Shannon H' index, GAMs, Pearson correlations, linear regression on residuals, etc. For instance, "GAMs were fit using the function 'gam (y~s, method = "REML")' from the mgcv package v1.9-1 (Wood, 2017)." Furthermore, the interpretations and discussion rely heavily on Pearson correlations – please add some justification for why this method was chosen over others, i.e., Spearman rank-correlations. For datasets that follow a bivariate normal distribution, Pearson correlations are useful to measure linear relationships (not sure if you have tested for whether your datasets are normally distributed). However, if the datasets are nonnormally distributed or have relevant outliers, you might actually consider using an alternative like Spearman correlation to test for monotonic association. This could provide different interpretations, potentially stronger correlations, than what your current results indicate.

As suggested, we now include details for the different functions used. The added text is:

- Line 243-244: "The Shannon Diversity Index was calculated for each group with the QIIME2 diversity plugin."
- Lines 312-317: "All correlations and models were generated with R v4.3.2. Specifically, Pearson correlations were performed with the function cor.test(x, y, method = "pearson"). Linear regressions were performed with the function lm(y~x), and residuals from the linear models were calculated with the resid() function. Generalized additive models (GAMs) were fit using the function gam(y~x, method = "REML) from the mgcv package v 1.9-1 (Wood, 2017)."

With regards to our choice to use Pearson correlations instead of Spearman correlations, Pearson correlations are useful to assess linear relationships as described by the reviewer. If we consider both a single organism which would have a fixed DNA copy number and a lack of variation in pigment quantities per cell from environmental conditions, the relationship between DNA and pigments ideally should be linear. Therefore, we specifically chose to use Pearson correlations to test the strength of this hypothesized linear relationship. Although a Spearman correlation may show stronger correlations, it does not necessarily help test this hypothesis. Pearson correlations have also been used in similar published analyses such as:

- Alexandra E Jones-Kellett, Jesse C McNichol, Yubin Raut, Kelsy R Cain, François Ribalet, E Virginia Armbrust, Michael J Follows, Jed A Fuhrman, Amplicon sequencing with internal standards yields accurate picocyanobacteria cell abundances as validated with flow cytometry, *ISME Communications*, Volume 4, Issue 1, January 2024, ycae115, https://doi.org/10.1093/ismeco/ycae115
- Qicheng Bei, Nathan L R Williams, Laura E Furtado, Daria Di Blasi, Jelani Williams, Vanda Brotas, Glen Tarran, Andrew P Rees, Chris Bowler, Jed A Fuhrman, Quantitative metagenomics for marine prokaryotes and photosynthetic eukaryotes, *ISME Communications*, 2025;, ycaf131, https://doi.org/10.1093/ismeco/ycaf131
- Lin Y, Gifford S, Ducklow H, Schofield O, Cassar N2019.Towards Quantitative Microbiome Community Profiling Using Internal Standards. Appl Environ Microbiol 85:e02634-18.https://doi.org/10.1128/AEM.02634-18
- Catlett, D., Siegel, D.A., Matson, P.G., Wear, E.K., Carlson, C.A., Lankiewicz, T.S. and Iglesias-Rodriguez, M.D. (2023), Integrating phytoplankton pigment and DNA meta-barcoding

observations to determine phytoplankton composition in the coastal ocean. Limnol Oceanogr, 68: 361-376. https://doi.org/10.1002/lno.12274

**Results and Discussion**

Lines 303 – 307: This section discussing the results of the cyanobacteria fraction of the data could be expanded a bit more. For instance, this potential dominance of *Prochlorococcus* might align with the observed warming influence and advection of oligotrophic offshore waters into the study region as previously observed at the San Pedro Ocean Time-series (SPOT) where this was accompanied by a notable shift from cold-water ecotypes to warm-water ecotypes during 2014-2015 (Yeh and Fuhrman, 2022 - https://doi.org/10.1038/s41467-022-35551-4). Similarly, the 2015-2016 El Niño event also marked an increase in an open ocean ecotype of UCYN-A at SPOT (Fletcher-Hoppe et al., 2023 - https://doi.org/10.1038/s43705-023-00268-y) but it seems its presence and range of coverage was not detected with the cyanobacteria ASVs recovered from the samples collected in this study.

> We now clarify that the cyanobacterial community overwhelmingly comprised *Procholorococcus* and *Synechococcus* with only extremely minor contributions from unclassified cyanobacteria or cyanobacterial diazotrophs on lines 347-349:
>
> > "On average, *Prochlorococcus* and *Synechococcus* accounted for 99.2% of 16S reads, with minor contributions from ASVs that were not resolved to lower taxonomic levels or cyanobacterial diazotrophs such as *Richelia* and UCYN-A."
>
> We agree with the reviewer that it would be interesting to further examine patterns and drivers such as warming that lead to *Prochlorococcus* dominance. However, the goal of this study is to compare abundances among the different measurements; therefore, we believe that such an analysis is beyond the scope of this manuscript.

Figure 3: Consider specifying "All Cyanobacteria" on the figure's panel titles C and D to align with the description in the figure legend. And same thing for Figure S7.

> As suggested, we have changed these figure titles as well as the title in Figure S10 to "All Cyanobacteria." Figure S7 is now Figure S9.

Line 497: "…;however, contrary to predictions" Are there literary references to suggest that diversity and richness should be expected to be low in deep SCML samples – where/why did you have that hypothesis?

> The first paragraph of this section states, "marine phytoplankton are presumed to exhibit a unimodal distribution with maximum diversity at an intermediate level of productivity, including within models of phytoplankton communities in the California Current Ecosystem (Irigoien et al., 2004; Li, 2002; Goebel et al., 2013)." Thus, low productivity such as those in deep SCMLs is predicted to be associated with low diversity. We have rephrased this sentence to be clearer and now state on lines 560-564:
>
> > "As predicted, the deepest SCML samples displayed the lowest NPP rates; however, diversity and richness remained high in these samples resulting in an absence of the positive side of a

> unimodal distribution that phytoplankton communities are expected to display (Fig. S8) (Irigoien et al., 2004; Li, 2002; Goebel et al., 2013)."

Figure 5: Are the samples highlighted in panel F only a subset of the samples from panel E? It is specified that the samples are ordered by the associated fucoxanthin concentrations, but it seems that only samples above a certain *dabA* expression threshold are included here – maybe clarify this selection criteria.

> Yes. As the reviewer states, the samples in panel F only include samples from panel E where *dabA* expression was detected. As suggested, this is now clarified in the caption: "Relative abundances of *Pseudo-nitzschia* species from ITS2 sequencing (left y-axis) and total *dabA* expression (right y-axis) **for samples where *dabA* was detected**. Samples are ordered by fucoxanthin concentrations as shown in Panel E."

**Technical comments:**

Line 48: "Earth system**s** models" (make it as "system" – singular)

> Corrected.

Line 137: Station 81.8 46.9 – are these two separate stations or just a unique nomenclature?

> This is a unique identifier. CalCOFI stations are identified first by line number and station number; therefore, both numbers are required.

Line 481: "…for the mechanisms that underlying them." Awkward phrasing.

> We have corrected this sentence by removing "that."

Line 535: Adjust to "…shown to produce DA and its production is…" You already introduced the acronym DA to represent domoic acid so you can maintain consistency this way.

> We have corrected this text to use the abbreviation instead of "domoic acid."

Lines 542-544: Consider rephrasing the sentences to streamline the strucutre: "Dinoflagellates, including certain members in the genera *Alexandrium*, *Dinophysis*, and *Gonyaulax* and species *Gymnodinium catenatum* and *Lingulodinium polyedra*, may also cause HABs globally and in the region (Anderson et al., 2012, 2021; Trainer et al., 2010; Ternon et al., 2023)."

> We have revised this text, largely as suggested. It now reads on lines 626-628: "Some dinoflagellates, including certain members of the genera *Alexandrium, Dinophysis*, and *Gonyaulax* as well as the species *Gymnodinium catenatum* and *Lingulodinium polyedra*, may also cause HABs in this regions and others globally (Anderson et al., 2021; Trainer et al., 2010; Ternon et al., 2023; Anderson et al., 2012)."

Lines 545 – 546: "although 39% of V4 and 55% of V9 18S copies…" Wouldn't referencing Figure S6B better point to these percentages – not sure the reference to Figure S13 here? Also, does blasting those sequences improve the taxonomic resolution to help better assess if there are potentially more HAB species which may currently be unassigned as HABs due to insufficient taxonomic resolution?

Yes, we thank the reviewer for pointing out this error. We have modified this reference to now be Figure S6B.

To taxonomically annotate our ASVs, we employed a relatively conservative approach by using the naïve-Bayes classifier implemented in QIIME2 (Bokulich et al., 2018) and the $PR^2$ database which is curated (Guillou et al., 2012). While we may be able to generate additional taxonomic assignments by using BLAST with another database, such as those from NCBI, we believe this approach is highly likely to introduce false positives, particularly with the lack of curation and high potential for mis-annotated sequences. We prefer to take a more conservative approach, even if that means that there are a higher percentage of sequences without more detailed classification; therefore, we have not made further revisions in response to this comment.

**References**

Anderson, D. M., Cembella, A. D., and Hallegraeff, G. M.: Progress in understanding harmful algal blooms: paradigm shifts and new technologies for research, monitoring, and management, Annu Rev Mar Sci, 4, 143–176, 2012.

Anderson, D. M., Fensin, E., Gobler, C. J., Hoeglund, A. E., Hubbard, K. A., Kulis, D. M., Landsberg, J. H., Lefebvre, K. A., Provoost, P., and Richlen, M. L.: Marine harmful algal blooms (HABs) in the United States: History, current status and future trends, Harmful Algae, 102, 101975, 2021.

Bokulich, N. A., Kaehler, B. D., Rideout, J. R., Dillon, M., Bolyen, E., Knight, R., Huttley, G. A., and Gregory Caporaso, J.: Optimizing taxonomic classification of marker-gene amplicon sequences with QIIME 2's q2-feature-classifier plugin, Microbiome, 6, 90, 10.1186/s40168-018-0470-z, 2018.

Callahan, B. J., McMurdie, P. J., Rosen, M. J., Han, A. W., Johnson, A. J. A., and Holmes, S. P.: DADA2: High-resolution sample inference from Illumina amplicon data, Nat Meth, 13, 581–583, 10.1038/nmeth.3869, 2016.

Gloor, G. B., Macklaim, J. M., Pawlowsky-Glahn, V., and Egozcue, J. J.: Microbiome datasets are compositional: and this is not optional, Front Microbiol, 8, 2224, 2017.

Goebel, N., Edwards, C., Zehr, J., Follows, M., and Morgan, S.: Modeled phytoplankton diversity and productivity in the California Current System, Ecol Model, 264, 37–47, 2013.

Guillou, L., Bachar, D., Audic, S., Bass, D., Berney, C., Bittner, L., Boutte, C., Burgaud, G., de Vargas, C., Decelle, J., del Campo, J., Dolan, J. R., Dunthorn, M., Edvardsen, B., Holzmann, M., Kooistra, W. H. C. F., Lara, E., Le Bescot, N., Logares, R., Mahé, F., Massana, R., Montresor, M., Morard, R., Not, F., Pawlowski, J., Probert, I., Sauvadet, A.-L., Siano, R., Stoeck, T., Vaulot, D., Zimmermann, P., and Christen, R.: The Protist Ribosomal Reference database (PR2): a catalog of unicellular eukaryote Small Sub-Unit rRNA sequences with curated taxonomy, Nucleic Acids Res, 41, D597–D604, 10.1093/nar/gks1160, 2012.

Irigoien, X., Huisman, J., and Harris, R. P.: Global biodiversity patterns of marine phytoplankton and zooplankton, Nature, 429, 863–867, 10.1038/nature02593, 2004.

Li, W. K. W.: Macroecological patterns of phytoplankton in the northwestern North Atlantic Ocean, Nature, 419, 154–157, 10.1038/nature00994, 2002.

Lin, Y., Gifford, S., Ducklow, H., Schofield, O., and Cassar, N.: Towards Quantitative Microbiome Community Profiling Using Internal Standards, Appl Enviro Microbiol, 85, e02634–02618, 10.1128/aem.02634-18, 2019.

McNichol, J., Williams, N. L., Raut, Y., Carlson, C., Halewood, E. R., Turk-Kubo, K., Zehr, J. P., Rees, A. P., Tarran, G., and Gradoville, M. R.: Characterizing organisms from three domains of life with universal primers from throughout the global ocean, Scientific Data, 12, 1078, 2025.

Ternon, E., Carter, M. L., Cancelada, L., Lampe, R. H., Allen, A. E., Anderson, C. R., Prather, K. A., and Gerwick, W. H.: Yessotoxin production and aerosolization during the unprecedented red tide of 2020 in southern California, Elementa: Sci Anthropocene, 11, 2023.

Trainer, V. L., Pitcher, G., Reguera, B., and Smayda, T.: The distribution and impacts of harmful algal bloom species in eastern boundary upwelling systems, Prog Oceanogr, 85, 33–52, 2010.

Wood, S. N.: Generalized additive models: an introduction with R, chapman and hall/CRC2017.

Yeh, Y.-C., McNichol, J., Needham, D. M., Fichot, E. B., Berdjeb, L., and Fuhrman, J. A.: Comprehensive single-PCR 16S and 18S rRNA community analysis validated with mock communities, and estimation of sequencing bias against 18S, Environ Microbiol, 23, 3240–3250, https://doi.org/10.1111/1462-2920.15553, 2021.

---

## Author Response (AR2)

We thank the reviewers for their additional comments and help improving our manuscript. Please see our responses in line below.

**Referee #1**

The manuscript has shown substantial improvement in both structure and fluency compared to the initial version. All suggestions have been duly considered and appropriately addressed. As a final remark, I would recommend revising the concluding sentence of the abstract, which could benefit from reformulation. Overall, the manuscript is suitable for acceptance.

We have revised the concluding sentence of the abstract to now read:

"Altogether, these results suggest that potential models of pigment concentrations via hyperspectral remote sensing may enable improved assessments of global phytoplankton community structure. These assessments may further support the detection of harmful algal blooms and the development of Earth system models."

**Referee #2**

The authors have done a detailed revision to address many of the suggestions, feedback, and questions the reviewers had with the earlier version of the manuscript. In the latest version, I only found very minor typos that should be corrected:

Line 184: "HPL method as ..." correct to "HPLC".

The abbreviation HPL refers to "Horn Point Laboratory" rather than HPLC. It is necessary to use this abbreviation as that is how it is referred to in the cited reference that describes the method in detail (Hooker 2005). We now clarify this by stating: "until analysis at the Horn Point Analytical Services Laboratory (HPL) at the University of Maryland with the HPL method as described in Hooker (2005)"

Line 316/317: Remove extra period before the citation: "(Benjamini and Hochberg, 1995).

Line 424: "were elevated wat lower DNA abundances..." correct to "at".

Line 480 (Figure 3 Legend): "(E) Comparison between Prochloroccus cells..." correct to "Prochlorococcus"

Line 490: "While HPLC pigments are used validating remote-sensing..." correct to "are used for validating"

We have corrected all of these errors. Thank you.